# Regulation of corneal stromal cell behavior by modulating curvature using a hydraulically-controlled organ chip array

Minju Kim [1], Kanghoon Choi [1], David Križaj [2] & Jungkyu Kim [1,2] ✉

Corneal curvature abnormalities drive ectatic diseases, yet their mechanobiological effects on stromal cells remain poorly understood. We developed a hydraulically controlled curvature array chip recapitulating disease-relevant geometries (33-56D) to investigate how keratocytes, fibroblasts, and myofibroblasts respond to geometric stress. Curvature-induced mechanical stress triggered dramatic cellular remodeling keratocytes exhibited significant proliferative enhancement and phenotypic transformation with ALDH3A1 downregulation and α-SMA upregulation, indicating mechanobiologically driven fibrotic activation. Fibroblasts developed curvature-dependent orthogonal alignment that recapitulates native corneal lamellar organization without chemical cues, while myofibroblasts showed enhanced contractile responses. RNA sequencing revealed that geometric stress activates identical molecular pathways dysregulated in keratoconus, including TGF-β/SMAD signaling, ECM-receptor interactions, and inflammatory cascades. Extracellular matrix remodeling was cell-type specific, with keratocytes showing homeostatic control loss, fibroblasts promoting matrix deposition, and myofibroblasts driving degradation. These findings establish curvature-induced mechanotransduction as the fundamental driver of corneal ectatic disease progression, repositioning geometric stress from a passive consequence to an active determinant of pathology.

Corneal shape abnormalities arising from ectatic diseases such as keratoconus[1,2], cornea plana, and keratoglobus[3] profoundly alter corneal biomechanics and optical properties, yet their mechanobiological influence on the stromal layer remains poorly understood. These diseases affect approximately 1 in 2000 individuals[4,5], and are characterized by substantial deviations in corneal curvature. Healthy corneas typically exhibit a curvature of ~43 diopters (D), corresponding to a radius of 7.8 mm[6]. By contrast, cornea plana exhibits curvature below 36 D[7], while keratoconus and keratoglobus display with curvatures exceeding 45 D[8].

The corneal stroma, the thickest and most structurally dominant layer of the cornea, is particularly sensitive to mechanical deformation induced by abnormal curvature. Normally, it is organized into 200–300 orthogonally aligned collagen lamellae surrounded by small leucine-rich proteoglycans (SLRPs), which regulate collagen organization and fibrillogenesis[9,10]. Mechanical disruption of this organized architecture through geometric deformation fundamentally alters cellular behavior and triggers pathological remodeling cascades. Disruptions of SLRPs has been linked to altered curvature and extracellular matrix (ECM) remodeling, particularly in keratoconus[11]. This leads to excessive fibrotic responses and corneal haze, characteristic of vision loss in these diseases[12]. Consequently, proper regulation of ECM deposition and degradation is crucial for maintaining corneal shape and transparency.

[1]Department of Mechanical Engineering, University of Utah, Salt Lake City, USA. [2]Department of Ophthalmology and Visual Sciences, University of Utah, Salt Lake City, USA. ✉e-mail: jkim@mech.utah.edu

The alteration in ECM regulation is closely linked to cellular transdifferentiation during fibrosis. Basic fibroblast growth factor (bFGF) and transforming growth factor beta (TGF-β) are critical signaling messengers that modulate cell migration, phenotypic behavior, and ECM secretion by corneal keratocytes[13,14]. TGF-β not only drives the transformation of quiescent keratocytes into myofibroblasts but also promotes keratocyte differentiation into fibroblasts, while bFGF primarily induces keratocyte-to-fibroblast transition. In addition, other growth factors such as PDGF and serum components, together with extracellular matrix (ECM) environments, have been reported to influence phenotypic transformation[15]. Although these biochemical and ECM-mediated pathways are well characterized, the mechanobiological mechanisms underlying stromal fibrosis in response to corneal shape abnormalities remain largely unexplored.

Corneal curvature represents a fundamental geometric feature that traditional biomechanical models, such as uniaxial and biaxial stretching systems, cannot adequately reproduce. Unlike uniform stretch, curvature generates spatially heterogeneous mechanical cues inherent to the curved architecture. These cues generate region-specific mechanical environments that are not captured by flat or uniformly stretched substrates. Previous studies have shown that curvature can guide morphological changes, curvotactic cell migration, cell differentiation, and actin cytoskeleton reorganization. For example, adherent cells on curved surfaces preferentially migrate and position within concave valleys, driven by curvature-induced traction forces[16,17]. Studies employing microfabricated curvature substrates have demonstrated that closely linked millimeter-scale curvatures promote collective cell migration and the self-organization of structures involving lamellipodial and filopodial protrusions[18–20]. These findings suggest that curvature can act as a potent mechanical cue that directs cellular architecture and behavior. However, a comprehensive understanding of how disease-relevant curvatures regulate phenotypic transitions and ECM organization in corneal stromal cells is still lacking.

In this study, we utilized a microfabricated, hydraulically-controlled curvature array chip as a corneal ectasia model to investigate how pathologically relevant curvatures—mimicking cornea plana, keratoconus, and keratoglobus—affect corneal keratocytes, fibroblasts, and myofibroblasts. Unlike conventional uniaxial or biaxial stretching models, our platform generates spatially distinct mechanical microenvironments that closely replicate the mechanical environment of ectatic corneas. We characterized curvature-induced mechanical cues through finite element analysis and systematically evaluated cellular responses across multiple domains: proliferation, phenotypic alterations, ECM expression, focal adhesion formation, and mechanobiological signaling. This approach aims to provide a comprehensive corneal ectasia model that clarifies how disease-relevant curvature profiles regulate stromal cell phenotypes and matrix remodeling, offering mechanistic insights into corneal ectatic diseases and informing future therapeutic strategies.

## Results

### Design and characterization confirm reliable curvature generation in the array chip

The curvature array chip enables sustained, physiologically relevant deformation through precise hydraulic control. As described in Methods 4.1 and 4.2, the platform comprises a PDMS membrane overlaid on a hydraulic microfluidic chamber, controlled by injecting defined volumes of deionized water (DIW) (Fig. 1a, b). Material characterization revealed that the PDMS membrane exhibited an elastic modulus of approximately 530 kPa, with the polydopamine coating showing slightly increased stiffness (Supplementary Fig. 1). This elastic modulus falls within the physiological range of corneal tissue (10–641 kPa)[21], ensuring biomechanically relevant substrate properties.

Using side-view time-lapse imaging and quantitative angle analysis (Supplementary Fig. 2), three distinct curvature levels were precisely generated by injecting specific DIW volumes: 172 μL (low: 11.79°,

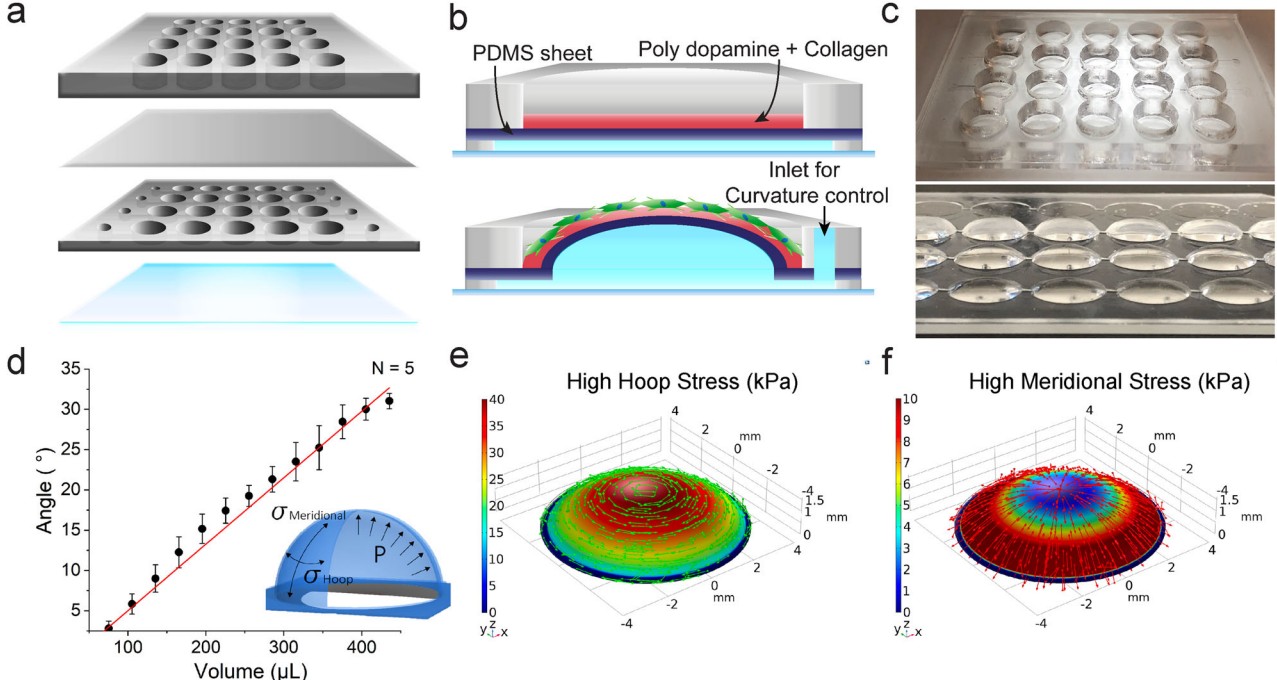

**Fig. 1 | Design and characterization of the hydraulically controlled curvature chip. a, b** Detailed curvature array chip, consisting of a medium reservoir, thin polydimethylsiloxane (PDMS), a hydraulic control layer, and a glass substrate from 1 to 4. By injecting water into inlets using a syringe, the desired curvature is formed by inflating a thin PDMS membrane. **c** Various curvatures by controlling the injection volume. **d** Relationship between volume and angle profiles during fluid injection into the hydraulic layer. Data are presented as mean ± standard deviation (s.d.), with N indicating the number of independent replicates (unit = chip). **e, f** Simulation results to present hoop and meridional stress profiles on high curvatures. Source data for panel d are provided as a Source Data file.

33.75 D), 212 μL (medium: 15°, 42.18 D), or 286 μL (high: 20.91°, 56.25 D), while a flat control was maintained with 45 μL until day 3 (Fig. 1c, d). The flat control chip was fabricated without a hydraulic chamber to provide a true zero-curvature reference.

To assess temporal stability, curvature was measured between day 0 and day 9 (see Supplementary Fig. 3 and Supplementary Note 1). Minimal curvature reductions were observed−1.03° (low), 0.35° (medium), and 0.32° (high)−corresponding to <8% loss in low curvature and <2% loss in medium and high curvature conditions (Supplementary Fig. 4). All curvatures remained within pathologically relevant curvature ranges over 9 days. The self-sealing property of PDMS at injection sites effectively prevented leakage, enabling long-term cell culture without curvature loss.

### Finite element analysis reveals stress distributions in the curvature array chip

FEA revealed curvature-specific stress distributions critical for understanding subsequent cellular responses, with hoop stress predominating in central regions and meridional stress increasing toward the periphery (Fig. 1e, f). Quantitative analysis across three defined regions−center (A1), slope (A2), and edge (A3)−revealed spatially distinct mechanical environments (Supplementary Figs. 5, 6 and Supplementary Note 2). Hoop stress dominated at the center (A1), while meridional stress exceeded hoop stress at the edge (A3). Most importantly, meridional-to-hoop stress ratios increased progressively from 5% at the center to 40−150% at the edge, creating mechanical gradients that became more pronounced with increasing curvature. These FEA-predicted stress gradients establish the mechanobiological framework underlying the spatial patterns of cellular behavior observed throughout this study. The predominant hoop stress in central regions (A1) provides the mechanical basis for cellular organization and proliferation patterns, while elevated meridional stress at edge regions (A3) creates the mechanical environment that drives directional alignment and focal adhesion formation. This FEA analysis demonstrates how geometric curvature generates spatially heterogeneous mechanical cues that translate directly into region-specific cellular responses through mechanotransduction.

### Cell proliferation is altered by curvature in the array chip platform

To promote ECM adhesion and achieve stable collagen coating under curved conditions, PDMS surfaces were pre-treated with 0.01% (w/v) polydopamine (PDA) after oxygen plasma activation, as optimized through titration experiments (Supplementary Fig. 7 and Supplementary Note 3). Corneal stromal cells were centrally seeded within each well of the PDA-treated, collagen-coated chips to evaluate how curvature-induced mechanical stress affects mechanosensitive proliferation, with keratocyte seeding density increased to compensate for their inherently lower proliferative capacity (Supplementary Fig. 8 and Supplementary Notes 4 and 5). Cell coverage contour analysis revealed progressive outward growth from the central seeding point across all curvature conditions, with all cell types achieving near-complete coverage of the 4 mm radius by day 9 (Fig. 2a). Curvature-generated mechanical stress induced different proliferative responses across cell phenotypes during the critical expansion period between day 3 and day 6, with keratocytes demonstrating the most pronounced mechanosensitive proliferative response - coverage rates increasing from near-zero on flat substrates to respectively at high curvature, but showed more modest relative increases from their $5.3 \pm 1.06$ mm²/day at high curvature ($p < 0.001$) (Fig. 2b). Fibroblasts and myofibroblasts exhibited higher absolute proliferation rates than keratocytes, reaching $9.77 \pm 1.1$ mm²/day and $10.9 \pm 1.25$ mm²/day flat controls. Fold-change analysis revealed that keratocytes exhibited the greatest curvature sensitivity, with progressive increases from low ($3.3 \pm 0.87$ fold) to medium ($4.11 \pm 0.66$-fold) to high curvature ($5.11 \pm 1.02$-fold)

conditions (Supplementary Fig. 9). Notably, all cell types demonstrated significant curvature-dependent increases in mechanosensitive proliferation (p < 0.001 for all comparisons), suggesting that geometric curvature serves as a critical regulator of corneal stromal cell proliferative responses through mechanotransduction pathways, with keratocytes showing the highest mechanosensitive responsiveness to curvature-generated stress gradients despite their traditionally quiescent phenotype.

### Curvature drives phenotypic transitions in stromal cells, particularly quiescent corneal keratocytes

The three corneal stromal cell types were characterized and validated before curvature application on day 3 (Supplementary Note 6). To investigate curvature-induced phenotypic changes, we assessed the expression of ALDH3A1 (a corneal crystallin and keratocyte marker) and α-SMA (a fibrotic activation marker) on day 6 and day 9. Under flat conditions, keratocytes maintained high ALDH3A1 expression and negligible α-SMA, consistent with their quiescent phenotype in conventional culture (Supplementary Fig. 10). However, curvature exposure induced pronounced phenotypic shifts in keratocytes, with decreased ALDH3A1 and elevated α-SMA by day 6 (Fig. 3a−c), indicating fibrotic activation. This trend continued through day 9 (Supplementary Fig. 11), with the most pronounced reduction in ALDH3A1 observed at day 6 and a more prominent upregulation of α-SMA at day 9. Corneal fibroblasts exhibited elevated ALDH3A1 and α-SMA expression only under high curvature at day 6, followed by reduced ALDH3A1 and sustained α-SMA upregulation across all curvature conditions by day 9 (Fig. 3a−c; Supplementary Fig. 11). Corneal myofibroblasts displayed a curvature-dependent increase in ALDH3A1, as quantified by immunostaining (Fig. 3b). However, ALDH3A1 levels remained much lower than in keratocytes and fibroblasts and were barely detectable in fluorescence images (Fig. 3a), indicating a limited role in phenotypic reversion. α-SMA expression remained high and was further enhanced under curvature, particularly at high curvature on day 6 and medium curvature on day 9 (Fig. 3a−c; Supplementary Fig. 11).

RT-qPCR analysis corroborated these immunostaining observations (Fig. 3d). In keratocytes, *ALDH3A1* was modestly elevated under low and medium curvature, but sharply decreased under high curvature. *α-SMA* expression increased approximately ten-fold across all curvature conditions, consistent with immunostaining results. In fibroblasts, *ALDH3A1* declined with increasing curvature, while *α-SMA* increased modestly. In myofibroblasts, *ALDH3A1* remained low and showed no substantial increase, whereas *α-SMA* rose in proportion to curvature level. Collectively, these results demonstrate that curvature exerts profound effects on stromal cell phenotype, with *α-SMA* serving as a sensitive indicator of curvature-induced mechanotransduction.

### Curvature alters focal adhesion distribution and mechanobiological marker expression

Phosphorylated FAK (pFAK) serves as a mechanosensitive marker of integrin signaling during dynamic processes such as migration, whereas vinculin stabilizes cell−matrix adhesions and is typically associated with static mechanical cues[22,23]. On day 6, both vinculin and pFAK expression increased with curvature across all stromal cell types (Supplementary Fig. 12), indicating an early mechanoadaptive response. By day 9, expression of both markers declined under high curvature in keratocytes and fibroblasts, suggesting reduced focal adhesion stability or downregulated mechanotransduction under prolonged mechanical stress, despite intact F-actin architecture and no evidence of cell detachment. In keratocytes, vinculin localization shifted toward the perinuclear region under curvature, indicating reduced adhesion and potential migratory activation. Notably, pFAK levels in keratocytes were maintained under curvature conditions, reflecting continued mechanosensing despite reduced adhesion,

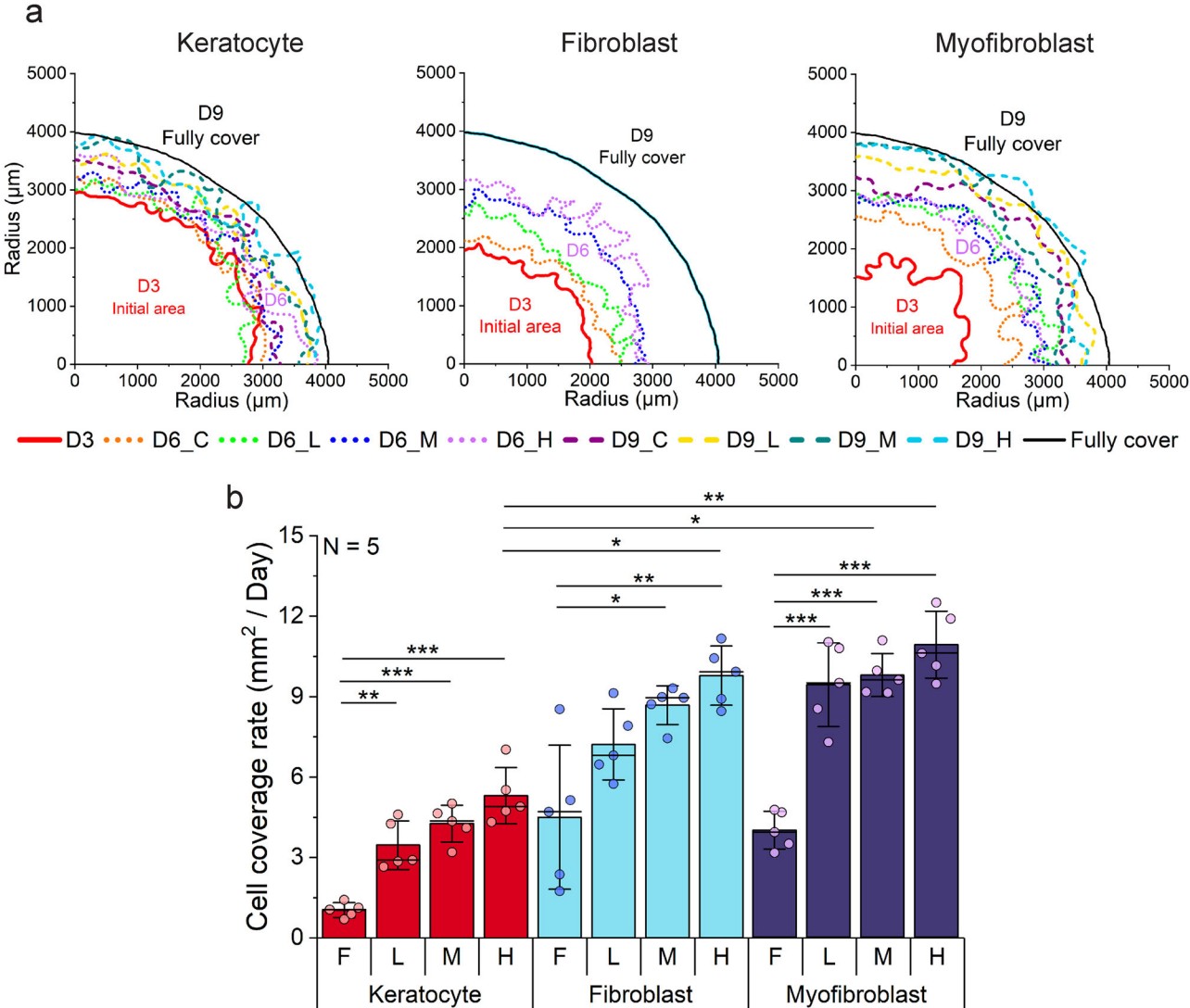

**Fig. 2 | Cell coverage rate of corneal stromal cells on curvature. a** Cell coverage contours of three corneal stroma cells on flat, low, medium, and high curvature. All cells achieved almost full coverage within six days from the starting point (day 3) **b** The cell coverage rates of cells on different curvatures were assessed between day 3 (D3) and day 6 (D6). All three cell types exhibited increased cell coverage rates on curvature. Specifically, fibroblasts and myofibroblasts displayed a faster cell coverage rate than keratocytes. Bar plots indicate mean ± standard deviation (s.d.), with ● representing individual data points (N = number of independent biological replicates, unit = chip). Statistical significance was assessed by two-sided one-way analysis of variance (ANOVA) followed by Scheffé's post hoc test (* $P < 0.05$, **$P < 0.01$, ***$P < 0.001$).

consistent with early migratory or EMT-like states. In contrast, myofibroblasts exhibited elevated vinculin and relatively low pFAK under curvature, consistent with their strong, stable adhesions and contractile phenotype. Quantitative analyses (Fig. 4, Supplementary Fig. 13) revealed curvature- and time-dependent regulation of vinculin. On day 6, vinculin increased in keratocytes and fibroblasts with curvature but declined by day 9. This decrease likely reflects phenotypic transitions with reduced cell-substrate adhesion[24,25]. Myofibroblasts maintained high vinculin expression across all curvature levels and time points, underscoring their contractile, adhesive nature.

Spatial profiling of vinculin across the center (A1), slope (A2), and edge (A3) regions closely matched FEA-predicted stress gradients (Supplementary Figs. 5–6). In keratocytes and fibroblasts (except fibroblasts under high curvature), vinculin expression increased from center (A1) to edge (A3) (Fig. 4b). However, under high curvature, fibroblasts displayed uniform expression across regions. Myofibroblasts showed opposite trends at low and medium curvature, with the highest vinculin levels in central regions, and uniform expression at high curvature, mirroring fibroblast behavior. These findings suggest

that elevated meridional stress relative to hoop stress at the edge (A3) under high curvature alters focal adhesion patterns in fibroblasts and myofibroblasts, promoting alignment. Thus, curvature guides spatial redistribution of focal adhesions, especially as cells transition from keratocytes to more contractile phenotypes. Enhanced edge adhesion and randomization may reflect collective migration behaviors observed on flat substrates, while higher curvature directs focal adhesion alignment along predicted stress vectors, thereby influencing cytoskeletal organization and establishing migratory polarity.

**Meridional and hoop stress profiles explain curvature-dependent alignment of stromal cells**

To complement vinculin localization findings, we analyzed stromal cell orientation patterns across curvature conditions and spatial regions. Cell alignment was significantly influenced by both curvature magnitude and spatial position along the curved surface. As shown in Fig. 5, orientation patterns varied across the center, slope, and edge, while Supplementary Fig. 14 summarizes overall orientation patterns across cell types and curvatures. On flat and low-curvature surfaces, cells

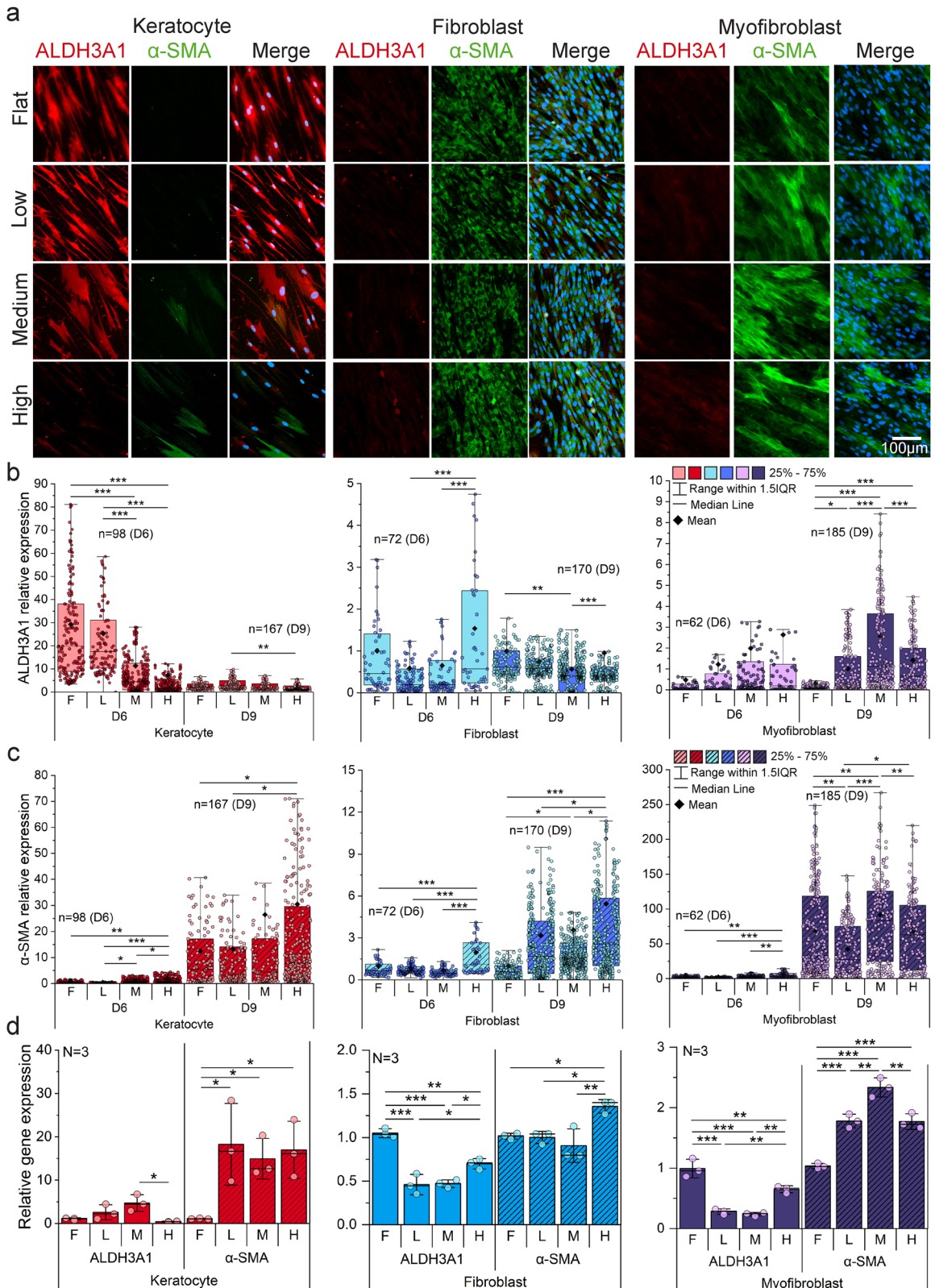

exhibited broad, disorganized angular distributions without preferential orientation, indicating random alignment (Fig. 5; Supplementary Fig. 14a–b). In contrast, medium and high curvature promoted preferential alignment across all cell types. Notably, corneal fibroblasts exhibited orthogonal actin fiber alignment under medium and high curvature conditions, closely resembling the characteristic lamellar architecture of the native anterior stroma. Tangent–vertical angles (θ$_{cross}$) at the orthogonal crossing point show 99.99° ± 3.73° at

medium curvature and 88.66° ± 1.65° at high curvature (n ≥ 5), confirming alignment near perpendicular orientation (Fig. 5a). To further quantify alignment efficiency, we calculated the proportion of fibroblasts aligned within ±10° of the orthogonal axis. Medium curvature yielded 48.4% efficiency, whereas high curvature showed 23.2%. This orthogonal organization was spatially dependent, occurring predominantly in the slope region rather than at the central peak, consistent with FEA-predicted stress distributions (Supplementary

**Fig. 3 | Phenotypic transformation of corneal stromal cells on curvature.**
**a** Immunofluorescence staining images of all cell types show the transforming cell phenotypes on curvature compared to flat using ALDH3A1 (red) and α-SMA (green), with nuclei counterstained by Hoechst (blue), on day 6 (3 days after curvature). **b, c** Quantitative mean intensity of ALDH3A1 and α-SMA on day 6 (D6) and day 9 (D9). ALDH3A1 expression in myofibroblasts gradually increases compared to flat, and the expression level of α-SMA on D9 was consistently higher on curvature for all cell types. Box plots indicate the 25th–75th percentiles (box), with whiskers extending to the minimum and maximum values within 1.5× the interquartile range (IQR). The center line denotes the median, ◆ indicates the mean, and ● represent individual data points, n = number of analyzed images. Statistical significance was assessed by two-sided one-way ANOVA followed by Scheffé's post hoc test (*P < 0.05, **P < 0.01, ***P < 0.001). **d** The expression levels of cell phenotype marker genes (ALDH3A1 and α-SMA) were analyzed with different curvatures and cell types using real-time quantitative reverse transcription polymerase chain reaction on D9 (6 days after applying the curvature), and all samples were quantified using the 2-ΔΔCt method compared to the flat. Bar plots indicate mean ± standard deviation (s.d.), with ● representing individual data points (N = independent biological replicates). Statistical significance was assessed by two-sided one-way analysis of variance (ANOVA) followed by Scheffé's post hoc test (* P < 0.05, **P < 0.01, ***P < 0.001). Source data are provided as a Source Data file.

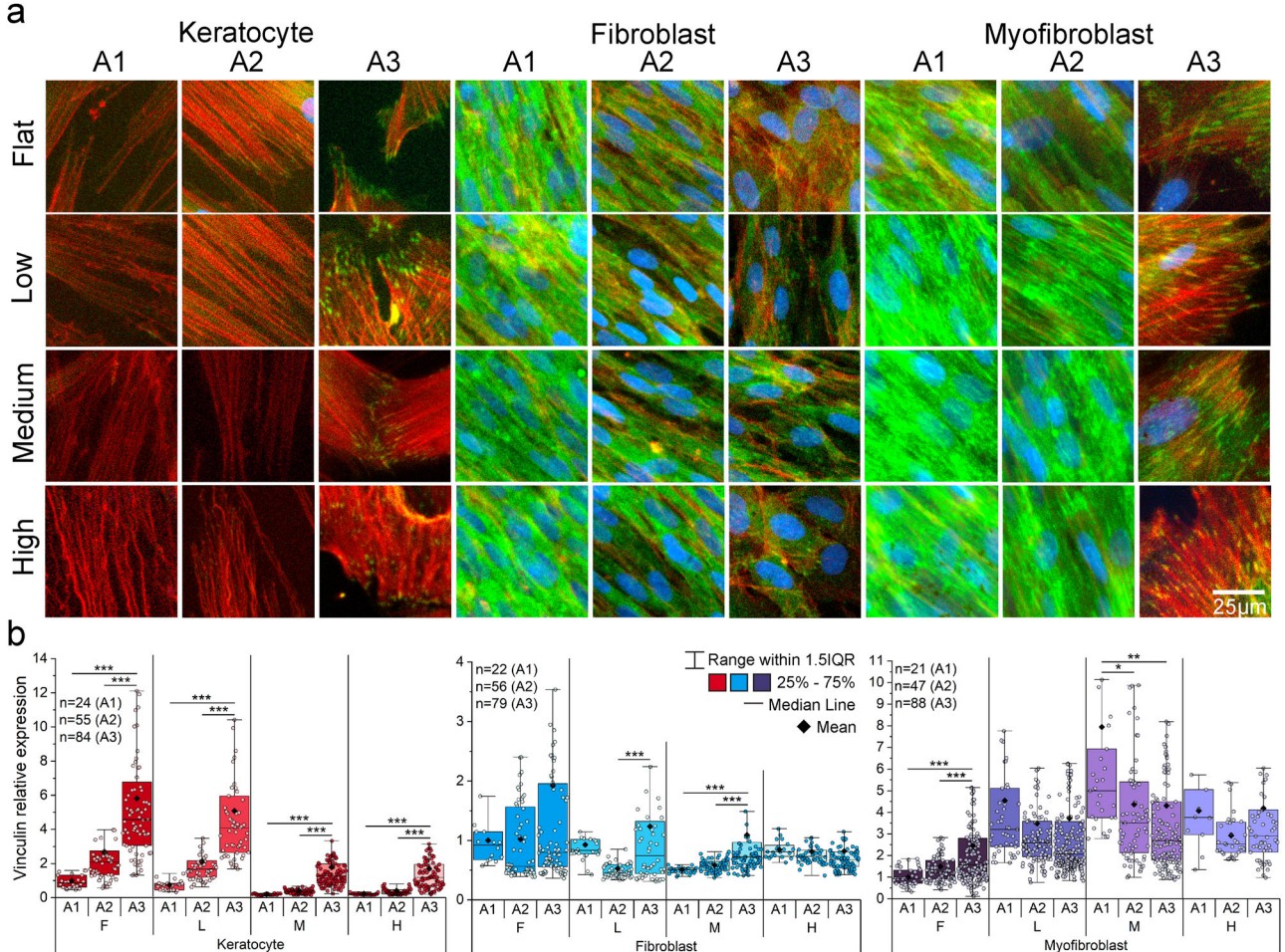

**Fig. 4 | Focal adhesion of corneal stromal cells on curvature. a** Vinculin stained 40 × magnification images of all cell types on day 9 showed a significant difference between slope (A2) and edge (A3) location. The expression on edge (A3) location for all cell types showed the increase of vinculin compared to the center (Green: Vinculin, Red: F-actin, Blue: Nuclei). **b** The quantitative expression of vinculin is shown in three locations (A1: center, A2: slope, A3: edge). This trend was distinct in cornea keratocytes, but myofibroblast showed the opposite trend on curvature. Box plots indicate the 25th–75th percentiles (box), with whiskers extending to the minimum and maximum values within 1.5× the interquartile range (IQR). The center line denotes the median, ◆ indicates the mean, and ● represent individual data points (n = number of analyzed images). Statistical significance was assessed by two-sided one-way analysis of variance (ANOVA) followed by Scheffé's post hoc test (*P < 0.05, **P < 0.01, ***P < 0.001), Source data for panel d are provided as a Source Data file.

Fig. 6c). Global cell orientation analysis (Fig. 5b), in which individual cell elongation axes were normalized to −90° to +90°, revealed region-specific differences across the center, slope, and edge regions. Keratocytes exhibited pronounced alignment variation between the center and edge under high curvature, whereas fibroblasts and myofibroblasts showed more uniform patterns. These alignment patterns closely paralleled regional vinculin expression profiles (Fig. 4b), indicating that focal adhesion dynamics directly contribute to curvature-guided cytoskeletal organization.

## Curvature regulates ECM expression patterns in stromal cells

Curvature significantly altered the expression of key ECM components across all corneal stromal cell types, revealing distinct, cell-type-specific mechanobiological responses. qRT-PCR analysis of collagens and small leucine-rich proteoglycans (SLRPs) revealed distinct patterns of matrix regulation dependent on both curvature level and cellular phenotype (Fig. 6). In keratocytes (Fig. 6a), *collagen types I, III,* and *V,* as well as SLRPs such as *lumican* and *keratocan,* were generally upregulated with increasing curvature. Notably, *collagen type I*

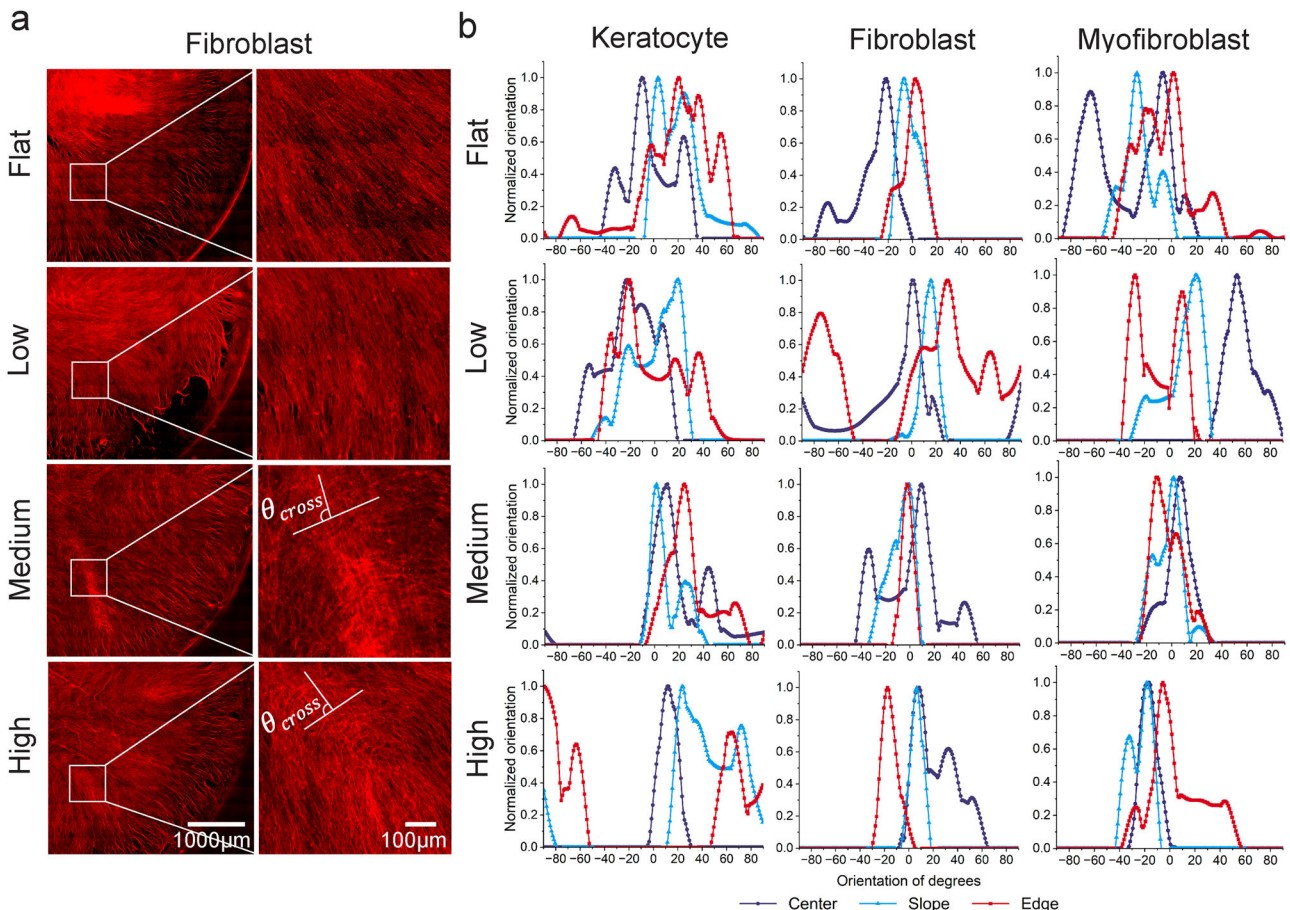

**Fig. 5 | Cell alignment of corneal stromal cells on curvature. a** F-actin tile images of fibroblasts showing organized orthogonal alignment patterns under medium and high curvature compared to flat surfaces. Tangent-vertical angles ($\theta_{cross}$) at the orthogonal crossing points show near-orthogonal alignment. Representative images from three independent experiments with similar results are shown. **b** Global cell orientation analysis across the center, slope, and edge regions. Each cell's elongation axis was measured, and orientations were normalized to $-90°$ to $+90°$. Sharper distributions around 0° indicate stronger alignment of cells in a uniform direction. Source data for panel b are provided as a Source Data file.

decreased sharply under high curvature, and *lumican* showed a bell-shaped response, peaking at medium curvature before declining at high curvature. These findings suggest that excessive curvature disrupts ECM homeostasis in keratocytes.

In contrast, fibroblasts and myofibroblasts showed progressive increases in *collagen types I, III*, and *V* under medium and high curvatures. At low curvature, fibroblast *collagen III* and myofibroblast *collagen I* and *V* remained similar to flat controls. Compared to keratocytes, activated cell types (fibroblasts and myofibroblasts) exhibited more moderate curvature-induced changes in ECM gene expression, indicating that keratocytes are more sensitive to curvature-mediated mechanotransduction. SLRP expression further distinguished activated cell types. Fibroblasts showed a significant upregulation of *keratocan*, while *lumican* levels modestly increased starting at medium curvature. Myofibroblasts demonstrated a more pronounced response, with increased expression of both *lumican* and *keratocan* at medium curvature; however, *keratocan* levels declined significantly at high curvature. These differential ECM responses have significant implications for understanding corneal pathophysiology. Downregulation of *lumican* and *keratocan* in keratocytes under high curvature may contribute to corneal opacity and collagen disorganization, as these SLRPs regulate fibril spacing and organization (Fig. 6a). Simultaneously, the enhanced collagen production by activated stromal cells under high curvature conditions may drive the fibrotic remodeling characteristic of progressive keratoconus.

## Bulk RNA-sequencing reveals curvature-dependent transcriptional remodeling

To comprehensively characterize curvature-induced transcriptional changes and complement qRT-PCR findings, particularly the downregulated responses observed in keratocytes and myofibroblasts, bulk RNA sequencing was performed. Venn diagrams and principal component analysis (PCA) shown in Fig. 7a, b revealed clear segregation by both cell phenotype and curvature conditions. PCA showed distinct clustering across all samples (Fig. 7b), with keratocytes exhibiting the greatest separation along the first two principal components, indicating a strong transcriptional response to curvature. In contrast, fibroblast and myofibroblast samples clustered more tightly, suggesting more modest curvature-dependent shifts.

Hierarchical clustering of differentially expressed genes (DEGs) across all conditions revealed that curvature elicited cell type-specific transcriptional programs (Fig. 7c). Keratocytes demonstrated the strongest transcriptional response, exhibiting 3,779 unique DEGs under high curvature conditions, while fibroblasts and myofibroblasts showed 2,505 and 2,061 curvature-responsive genes, respectively. Detailed analysis of cell-type-specific DEGs revealed distinct response patterns. In keratocytes (Fig. 7d), volcano plot analysis showed significant upregulation of ECM remodeling genes such as *MMP23A* and *Fibrillin-3 (FBN3)*, while *Tenascin-C (TNC)* was consistently downregulated. Since *TNC* is a canonical marker of fibrotic ECM remodeling, its suppression indicates an alternative, potentially homeostatic matrix

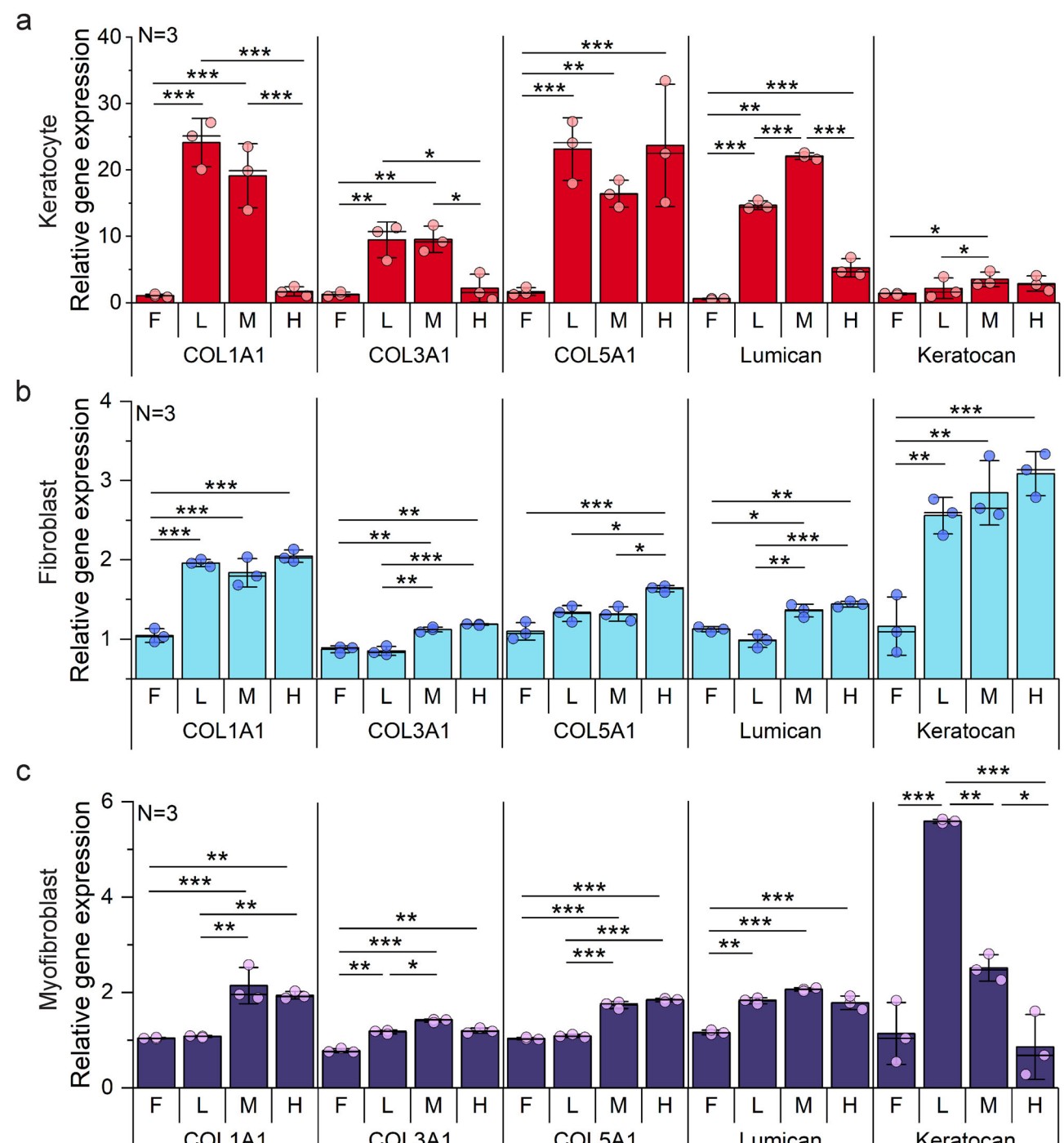

**Fig. 6 | Extracellular matrix (ECM)-related gene expression of corneal stromal cells on curvature.** Quantitative real-time reverse transcription polymerase chain reaction (RT-qPCR) analysis of ECM-related genes (*lumican, keratocan, COL1A1, COL3A1, COL5A1*) normalized to *GAPDH* using the 2−ΔΔCt method relative to flat conditions. **a** Keratocyte, **b** fibroblast, and **c** myofibroblast-specific expression profiles, Bar plots indicate mean ± standard deviation (s.d.), with ● representing individual data points (N = independent biological replicates). Statistical significance was assessed by two-sided one-way analysis of variance (ANOVA) followed by Scheffé's post hoc test (*$P < 0.05$, **$P < 0.01$, ***$P < 0.001$).

remodeling mechanism in keratocytes. Curvature also induced expression of mechanosensitive ion channels (Fig. 7e) and triggered TGF-β signaling changes, with *SMAD7* elevated at low curvature and *GDF6/GDF9* upregulated under high curvature. Inflammatory mediators, including *IL1B* and *TNF*, were also induced. Although the EMT marker *SNAI1* was upregulated, concurrent *TNC* downregulation suggests a controlled, partial EMT-like response rather than pathological fibrotic transformation. This intermediate state exhibits early profibrotic characteristics, including cytoskeletal reorganization, ECM

remodeling, and increased contractility, while preserving homeostatic controls that limit progression to irreversible fibrosis.

In fibroblasts (Fig. 7f), ECM remodeling genes including *LAMB4* and *MMP23B* were strongly induced. Heatmap analysis (Fig. 7g) showed progressive upregulation of *MMP12*, *COL6A4P2*, and *LAMB4* with increasing curvature, indicating active matrix deposition. Notably, *CDH1 (E-cadherin)* was markedly downregulated, suggesting altered adhesion dynamics. Multiple GPCR signaling and mechanotransduction-related genes were also upregulated. Induction

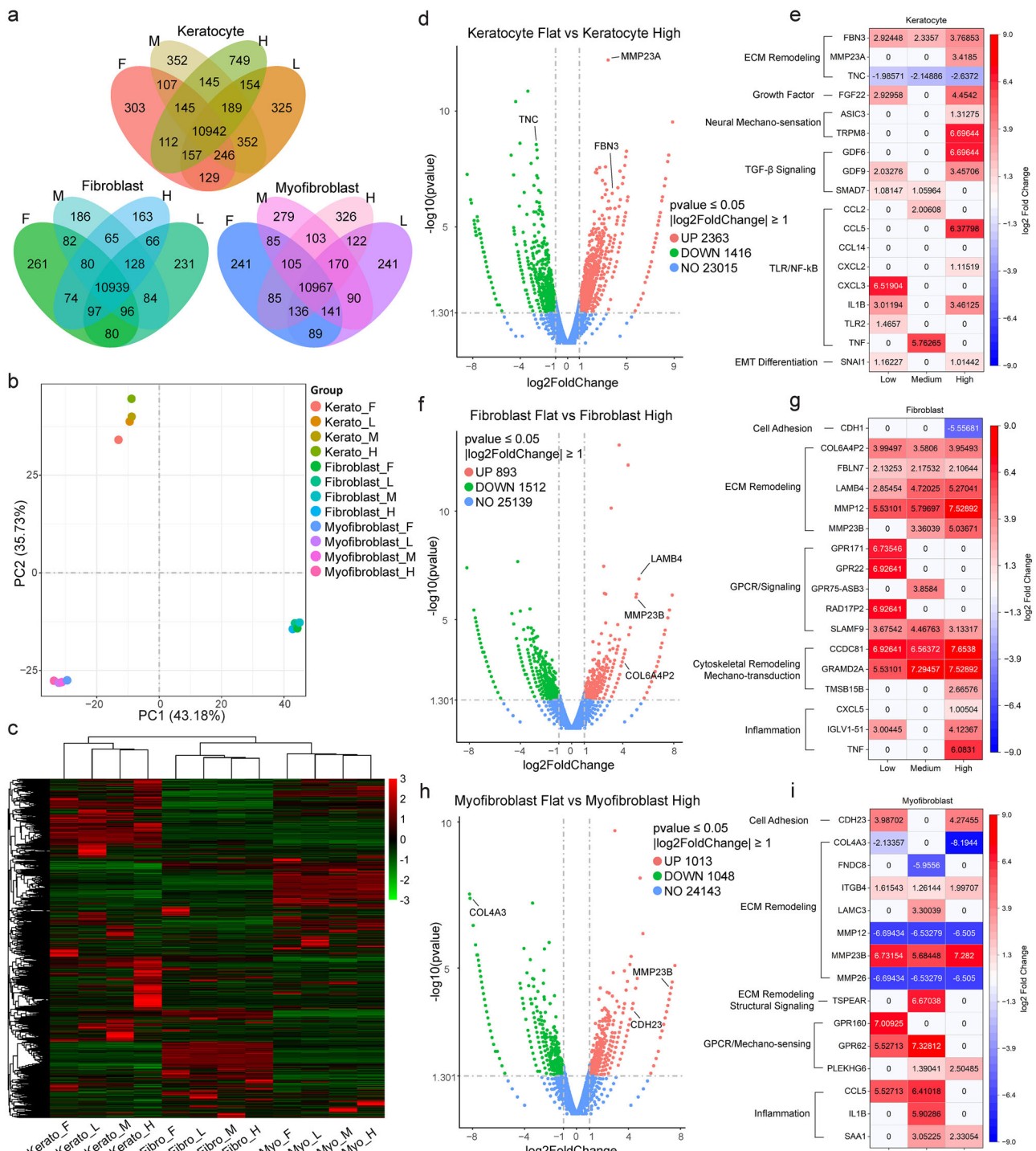

**Fig. 7 | Transcriptomic profiling of corneal stromal cells on curvature. a** Venn diagrams showing the number of shared and unique genes expressed under flat (F), low (L), medium (M), and high (H) curvature conditions in keratocytes, fibroblasts, and myofibroblasts. **b** Principal component analysis (PCA) reveals distinct transcriptomic separation by both cell phenotype and curvature level. Keratocytes exhibit the greatest dispersion, indicating a pronounced response to curvature. **c** Hierarchical clustering heatmap of all expressed genes illustrates cell type- and curvature-specific transcriptional profiles. **d, f, h** Volcano plots display

differentially expressed genes (DEGs) between flat and high curvature conditions in keratocytes (**d**), fibroblasts (**f**), and myofibroblasts (**h**). Genes with |log₂Fold-Change | ≥ 1 and *p*-value < 0.05 are considered significant and highlighted in red (upregulated) or green (downregulated). **e, g, i** Heatmaps showing curvature-dependent expression of selected genes involved in extracellular matrix (ECM) remodeling, mechanotransduction, and inflammation in keratocytes (**e**), fibroblasts (**g**), and myofibroblasts (**i**).

of inflammatory genes such as *TNF* at high curvature further supports this activated phenotype. Collectively, these findings demonstrate that fibroblasts show a robust curvature response characterized by enhanced ECM remodeling, altered cell adhesion, and pro-inflammatory signaling, consistent with fibrotic activation. In

myofibroblasts (Fig. 7h, i), *MMP23B* was consistently upregulated across all curvature conditions, while *MMP12* and *MMP26* were down-regulated, indicating a shift toward a degradation-dominant remodeling profile. *COL4A3* was significantly suppressed at high curvature. Mechanosensitive GPCRs including *GPR62* and *GPR160* were induced

at low to medium curvature, and inflammatory genes such as *CCL5*, *IL1B*, and *SAA1* were also elevated. These patterns indicate that myofibroblasts undergo matrix degradation and inflammation-mediated remodeling, potentially reflecting advanced fibrotic progression.

To further elucidate the biological processes and signaling pathways underlying these transcriptional responses, GO and KEGG enrichment analyses were performed by comparing flat and high curvature conditions across stromal phenotypes (Supplementary Figs. 15, 16). GO analysis revealed consistent enrichment of biological processes, including receptor signaling, ion transport, cytokine-mediated signaling, and extracellular matrix organization, with G-protein coupled receptor activity, cytokine activity, cation channel activity, and extracellular matrix emerging as recurrent features across all cell types. KEGG pathway analysis similarly identified shared enrichment of Notch signaling, PI3K-Akt signaling, MAPK signaling, ECM–receptor interaction, focal adhesion, and cytokine–cytokine receptor interaction. Furthermore, cell-type-specific analysis revealed distinct pathway priorities: keratocytes predominantly activated mechanotransduction and ion transport pathways, fibroblasts showed robust ECM remodeling and adhesion responses, while myofibroblasts exhibited pronounced wound healing and inflammatory activation. The consistent activation of keratoconus-associated pathways—including TGF-β/SMAD signaling, ECM-receptor interactions, and inflammatory responses—across all cell types indicates that pathological curvature drives disease-relevant molecular programs regardless of stromal cell phenotype. These findings demonstrate that curvature activates both shared and cell-type-specific transcriptional programs governing mechanotransduction, inflammation, and ECM remodeling, which directly overlap with pathways dysregulated in corneal ectatic diseases.

## Discussion

This study demonstrates that corneal curvature-induced mechanical stress serves as the primary driver of stromal cell remodeling, influencing proliferation, migration, phenotype, alignment, and ECM regulation. Unlike conventional biochemical models of corneal pathology, our findings identify geometry-induced stress as an upstream regulator that activates the molecular cascades traditionally attributed to chemical stimuli. Unlike conventional uniaxial or biaxial stretch models[26,27], curvature generates spatially heterogeneous mechanical environments dictated by substrate geometry. Finite element analysis revealed that curved substrates exhibit two principal stress components, meridional and hoop stresses, which vary across regions. Meridional stress dominates along the curvature slope and edge, promoting axial cell elongation and traction generation, while hoop stress becomes prominent near the apex, influencing circumferential alignment. These spatially resolved mechanical cues likely underlie the distinct alignment and migration patterns observed in stromal cells, reinforcing the role of curvature as a topographically encoded mechanical regulator.

The mechanosensitive proliferation profiles clearly demonstrate that curvature-induced mechanical stress serves as a potent regulator of corneal stromal cell behavior through distinct mechanotransduction pathways. Significant increase in keratocyte proliferation from flat to high curvature conditions indicates that these traditionally quiescent cells possess highly sensitive mechanosensing machinery that responds to geometric stress amplification. This pronounced mechanosensitive response likely involves integrin-mediated focal adhesion signaling and downstream activation of stress-responsive transcriptional programs that drive proliferative gene expression under mechanical loading. Previous studies using peptide amphiphilic (PA)-coated surfaces have shown that stromal cells migrate from the periphery to the apex of the curvature, forming aligned cell patterns[28]. Without PA coating, the cells were unable to migrate along the energetically unfavorable gradient of the curved substrate[29]. However, our findings reveal that stromal cells demonstrate intrinsic mechanosensitive capabilities that enable both proliferative responses and directional migration toward stress-concentrated regions without requiring specialized PA coating, using only standard adhesive surface preparation. This mechanosensitive proliferation represents a fundamental cellular adaptation mechanism where geometric cues trigger coordinated proliferative and migratory responses, supporting the concept that corneal stromal cells can perceive and respond to curvature-induced mechanical environments through evolutionarily conserved mechanotransduction pathways. Especially, the difference in keratocyte proliferative behavior, exhibiting the highest curvature sensitivity among all stromal cell types, suggested that these cells undergo fundamental changes beyond simple proliferative enhancement.

Such pronounced mechanosensitive responses likely reflect concurrent phenotypic alterations driven by the curvature-induced mechanical stress. Indeed, we observed that curvature alone, without chemical stimulation, was sufficient to induce phenotypic transformation in corneal stromal cells[30]. Quiescent keratocytes, typically non-contractile and characterized by high ALDH3A1 and low α-SMA expression, exhibited a marked reduction in ALDH3A1 and a concomitant upregulation of α-SMA in response to increasing curvature, indicating a shift toward a contractile, myofibroblast-like state. This transition was confirmed through both immunostaining and qRT-PCR analysis, demonstrating that mechanical cues alone are sufficient to trigger phenotypic alteration. To elucidate the molecular mechanisms underlying this transition, we performed bulk RNA sequencing. Among all stromal phenotypes, keratocytes showed the strongest transcriptional response to curvature, highlighting their unique mechanosensitivity. Several distinct but converging pathways were implicated in this transition. Upregulation of *SNAI1*, a master regulator of epithelial-to-mesenchymal transition (EMT), suggests engagement of EMT-like transcriptional programs, consistent with the spatially heterogeneous α-SMA expression observed at the protein level. TGF-β signaling was prominently modulated by curvature: *SMAD7*, a negative regulator of TGF-β/Smad signaling, was upregulated under low and medium curvature, indicating early feedback control, whereas *GDF6* and *GDF9*, pro-fibrotic ligands of the TGF-β superfamily, were strongly induced under high curvature, reflecting activation of downstream fibrotic signaling cascades. These findings support a model wherein curvature-dependent TGF-β activation is tightly regulated by mechanical stress intensity, mediating either controlled homeostatic responses or progressive phenotypic shifts. In parallel, enrichment of G protein-coupled receptor (GPCR) signaling genes, including *FGF22*, indicates that curvature is sensed through GPCR-mediated mechanotransduction, which likely acts upstream or in concert with TGF-β and Rho/ROCK signaling pathways. Moreover, upregulation of inflammatory mediators such as IL1B and TNF under curvature suggests that fibrotic transition is accompanied by a pro-inflammatory transcriptional program, further amplifying the remodeling response. Similar curvature-dependent transcriptional responses were observed in fibroblasts and myofibroblasts. However, these cell types exhibited distinct remodeling signatures, including elevated GPCR activity and activation of necroptosis-related pathways under high curvature, as revealed by GO term, KEGG. These findings suggest that curvature not only induces phenotypic transitions but also promotes pathological remodeling reminiscent of stromal alterations observed in corneal ectasia. Collectively, our results establish curvature as a potent mechanical regulator of stromal cell fate, activating a multi-pathway mechanotransductive cascade involving EMT, TGF-β signaling, GPCR-mediated mechanosensing, and inflammatory responses. This cascade drives keratocytes and to varying degrees, fibroblasts and myofibroblasts toward fibrotic states in a curvature-dependent manner, providing mechanistic insight into the role of geometry in corneal homeostasis and disease progression.

The contractile properties of corneal stroma cells are closely linked to the expressions of α-SMA[31] and the development of focal adhesions[32,33]. Keratocytes and fibroblasts exhibited dynamic adhesion remodeling in response to curvature, distinct from myofibroblasts. At early time points (day 6), a modest increase in α-SMA coincided with elevated vinculin levels, indicating active focal adhesion formation. However, by day 9, further α-SMA upregulation was accompanied by reduced vinculin, particularly in keratocytes, consistent with phenotypic transition and increased motility[34]. In contrast, myofibroblasts exhibited a parallel increase in both α-SMA and vinculin under curvature, reflecting their inherently contractile and adhesion-dependent phenotype. Integrin-mediated mechanotransduction was evident through early activation of phosphorylated FAK (pFAK) across all cell types[35,36]. Over time, pFAK levels declined in parallel with vinculin, although keratocytes maintained relatively sustained pFAK expression despite reduced structural adhesions. This sustained FAK signaling, combined with α-SMA and *SNAI1* upregulation, supports an EMT-like transition under prolonged curvature-induced mechanical stress[37]. Conversely, myofibroblasts showed stronger vinculin expression relative to pFAK, reinforcing their mature contractile identity. Focal adhesion dynamics were further linked to activity of Rho GTPases and the Arp2/3 complex, known mechanosensors that orchestrate cell migration, shape, and adhesion[38]. RhoA promotes filopodia and traction force generation in leading cells, while Arp2/3 drives lamellipodia formation and intercellular sensing in follower cells[39]. In Supplementary Fig. 17, Keratocytes showed reduced RhoA expression on curved substrates, aligning with decreased vinculin and suggesting low traction force and increased cell–cell interaction. Fibroblasts exhibited curvature-enhanced expression of both RhoA and ARPC2, especially at medium curvature, indicating a mixed migratory strategy balancing substrate adhesion and neighbor sensing. Myofibroblasts showed high RhoA and vinculin but low ARPC2, consistent with unidirectional, substrate-dominant migration. These mechanosensitive programs were corroborated by spatial localization of focal adhesions, indicating that mechanical stress gradients guide adhesion patterning and cytoskeletal alignment. Taken together, these findings indicate that curvature not only induces phenotypic transitions in stromal cells, but also orchestrates their adhesion signaling and spatial organization through integrated regulation of α-SMA, pFAK, Rho GTPase activity, and focal adhesion distribution. These coordinated responses enable phenotype-specific alignment and migration strategies, underlying distinct modes of mechanical adaptation to substrate geometry.

Cell alignment in response to curvature is tightly regulated by local mechanical environments, particularly spatial gradients in displacement and stress. On curved substrates, slope regions with high displacement gradients promote traction force generation[40], while edge regions with elevated stress concentrations enhance stress fiber formation[41]. These mechanical cues generate distinct meridional and hoop stresses[42,43], which together guide cytoskeletal organization. Across all stromal cell types, curvature-induced alignment in the slope direction, reflecting cellular responsiveness to directional mechanical cues. Among these, fibroblasts exhibited the most pronounced alignment response. Under low curvature, where hoop stress dominates in the absence of significant meridional tension, fibroblasts showed disorganized, random orientation, likely due to insufficient directional guidance. In contrast, medium and high curvature conditions generated stronger traction forces and induced spatial coordination between hoop and meridional stresses. This curvature-induced stress guided fibroblasts to align preferentially along the hoop axis, resulting in well-organized orthogonal alignment patterns that remained stable during long-term culture (4 weeks), supporting the robustness of curvature-driven organization (Supplementary Fig. 20). Interestingly, orthogonal alignment was more pronounced under medium curvature. Although high curvature presents a steeper slope, the geometrically accessible region available for cell alignment is broader in medium curvature, allowing a greater proportion of cells to organize orthogonally. Unlike previous studies requiring chemical induction or micropatterned substrates to achieve similar orthogonal patterns[44–47], our results demonstrate that curvature alone can drive orthogonal alignment within physiologically relevant geometry.

This orthogonal architecture may reflect fibroblasts' dual-mode mechanosensing, integrating cell–cell communication via Arp2/3-mediated lamellipodia with substrate anchoring through RhoA-driven filopodia and focal adhesions. Such integration may enable simultaneous responses to meridional and hoop stress vectors. Together, these results indicate that curvature-induced mechanical heterogeneity regulates not only focal adhesion dynamics and contractility, but also higher-order cytoskeletal architecture, resulting in distinct alignment strategies across stromal phenotypes.

The curvature-induced expression of ECM in stromal cells is consistent with their role in ECM deposition and remodeling. *Collagen types I* and *III* contribute to stromal structure and elasticity, while *type V* regulates the assembly of *type I* fibrils[48,49]. In fibroblasts and myofibroblasts, expression of these collagens progressively increased with curvature, consistent with the established paradigm that mechanical strain enhances ECM synthesis during tissue repair[50]. Conversely, keratocytes under high curvature conditions downregulated *collagen types I* and *III* despite increased expression of *type V*, suggesting suppression of primary matrix formation and activation of regulatory mechanisms to limit excessive ECM accumulation. Proteoglycan expressions were also curvature sensitive. *Lumican* and *keratocan*, which are essential for collagen fibril organization and stromal transparency, were downregulated under high curvature, particularly in keratocytes and myofibroblasts. *Lumican* deficiency is associated with enlarged, disorganized fibrils and corneal opacity[51,52], while *keratocan* knockout results in thinner corneas with compromised integrity[53,54]. These findings suggest that curvature-induced repression of proteoglycan expression contributes to matrix disorganization and reduced transparency, as seen in ectatic disease. Transcriptomic analysis further revealed distinct curvature-induced ECM remodeling programs across stromal cell types. In keratocytes, *tenascin-C (TNC)*, a matrix glycoprotein is involved in tissue remodeling, was consistently downregulated, while genes linked to partial epithelial–mesenchymal transition (EMT), such as *FBN3*, *MMP2*, and *SNAI1*, were upregulated. This indicates a shift toward homeostatic remodeling without overt fibrosis. Fibroblasts showed strong ECM accumulation signatures, including marked upregulation of *LAMB4* and *FBLN7*, indicative of active matrix deposition and organization. Elevated *MMP12* levels further imply a balanced remodeling environment during matrix expansion. In contrast, myofibroblasts exhibited a shift toward matrix degradation under high curvature. While low to medium curvature induced stabilizing genes such as *LAMC3* and *ITGB4*, high curvature selectively upregulated *MMP23B* and downregulated *MMP12*, indicating a phenotypic transition from matrix stabilization to degradation.

These cell-type-specific responses resemble the remodeling cascade characteristic of corneal ectatic diseases such as keratoconus, marked by progressive stromal thinning and shape deformation[55–57]. Under high curvature, keratocytes exhibited early disease-like features, including partial EMT and upregulation of pro-fibrotic mediators such as *SNAI1*, *GDF6*, and *GDF9*, suggesting loss of quiescence and initiation of matrix disorganization. Fibroblasts contributed to fibrotic ECM expansion, while myofibroblasts engaged in matrix degradation, reflecting advanced stages of ectatic remodeling. Together, these findings reveal a curvature-driven, stepwise remodeling process across stromal cell types, beginning with transcriptional shifts in keratocytes, followed by ECM accumulation in fibroblasts and degradation by myofibroblasts.

Abnormal curvature is closely associated with various corneal pathologies, particularly keratoconus—a non-inflammatory ectatic disorder characterized by stromal thinning and progressive anterior

bulging that can lead to visual distortion and, in severe cases, blindness[55–57]. As corneal ectasia progresses, keratocyte depletion and phenotype alteration renders the stroma increasingly vulnerable to mechanical stress and further deformation. A hallmark of this progression is the curvature-induced upregulation of collagen and α-SMA, both of which are key pathological markers associated with fibrotic remodeling[58]. Excessive deposition of collagen and proteoglycans alters the biomechanical properties of the stroma, reducing its resistance to dehydration and compromising its structural integrity[59]. This pathological shift is accompanied by enhanced focal adhesion formation and abnormal ECM deposition, both of which correlate with the extent of curvature deformation and disease severity in keratoconus[60,61]. Although keratoconus is not classified as an inflammatory disorder, studies have reported inflammatory responses in its post-onset stages[62,63]. Our findings align with these observations, suggesting that sustained mechanical stress induced by curvature deformation may initiate a secondary inflammatory response. This inflammation, in turn, likely amplifies ECM remodeling and exacerbates tissue dysfunction.

Our transcriptomic analysis revealed upregulation of RhoA/ROCK and TGF-β signaling pathways in response to substrate curvature, suggesting these as key mediators of mechanotransduction. However, a limitation of the current study is the absence of functional validation experiments, such as pharmacological inhibition or genetic knockdown approaches, to directly confirm the causal roles of these pathways in curvature-guided cellular behavior. These targeted perturbation studies, such as pharmacological inhibition or genetic knockout of the RhoA/ROCK pathway, could clarify whether curvature-induced phenotypic transitions are mechanistically governed by cytoskeletal tension and contractility[30]. Similarly, while our RNA sequencing results suggest activation of the TGF-β/Smad signaling pathway, further functional validation will be necessary to confirm its causal role in curvature-induced fibrosis. Beyond these pathways, future investigations should elucidate curvature effects at the molecular level, particularly the long-term impact of geometric deformation on corneal homeostasis, pathological progression, and wound healing responses. A critical area for exploration involves the molecular mechanisms underlying ECM deposition and degradation in response to corneal shape deformation, specifically through examining the activities of tissue inhibitors of metalloproteinase (TIMP) and matrix metalloproteinase (MMP)[64].

Our findings establish curvature-induced mechanotransduction as a fundamental driver of corneal ectatic disease progression. Across the clinically relevant curvature spectrum (33–56D), we demonstrated that geometric cues alone can drive cellular alignment that recapitulates the orthogonal lamellae organization of native corneal stromal architecture, proliferative enhancement, phenotypic transformation with α-SMA upregulation, altered matrix metalloproteinase expression profiles, and inflammatory gene upregulation. These findings reveal how geometric stress orchestrates both physiological organization and pathological remodeling in corneal ectatic diseases. Transcriptomic analysis further revealed that curvature-induced stress activates the same molecular pathways dysregulated in keratoconus, including TGF-β/SMAD signaling, ECM-receptor interactions, inflammatory cascades, and mechanosensitive ion channels. These insights reposition corneal curvature from being a passive consequence of pathology to an active determinant of disease advancement. These mechanistic insights directly validate geometric normalization as a therapeutic target. Curvature-adjusting implants, such as intracorneal ring segments, already demonstrate clinical efficacy by reshaping corneal geometry, improving visual acuity, and restoring corneal topography[65,66]. Building on these successes, future strategies may include bioengineered substrates or next-generation implants designed to modulate corneal geometry based on patient-specific mechanical stress patterns. Such approaches would directly leverage geometric stress modulation to alter cellular behavior and tissue remodeling at the root of disease progression. By identifying curvature-driven stress amplification as the central mechanistic force in corneal ectasia, our work provides both a unifying pathogenic model and a foundation for therapeutic innovation. Recognizing corneal geometry as not merely a marker but a mechanobiological driver of disease reframes ectasia management, guiding the development of mechanism-based interventions for earlier detection, prevention, and treatment of vision-threatening disorders.

## Methods

This study complies with all relevant ethical regulations. No human participants or animals were involved. Primary human corneal stromal cells were purchased from ScienCell Research Laboratories and used according to the manufacturer's protocols. The supplier confirmed that all cells were derived from donor tissue obtained under informed consent and ethical approval, and cells were authenticated and tested for mycoplasma contamination.

### Microfabrication of a hydraulically-controlled curvature array chip

The hydraulically-controlled curvature array chip was fabricated to replicate pathologically relevant corneal curvatures on a planar substrate. The chip comprised three main components: a hydraulic layer, a thin polydimethylsiloxane (PDMS) membrane (Supplementary Fig. 1), and a medium reservoir. Based on corneal anatomical data[63], a 4×5 array of 8 mm-diameter circular chambers was designed in AutoCAD (Autodesk, USA) (Supplementary Fig. 18) and cut from a 250 μm-thick silicone PDMS sheet (BISCO HT-6240, Rogers Corp., USA) using a vinyl cutter (GX-24, Roland DGA Corp., USA). For the membrane and reservoir layers, PDMS (SYLGARD 184, Dow Corning, USA) was prepared by mixing base and curing agent in a 10:1 ratio. After degassing under vacuum for 30 min, the mixture was poured into a 150 mm Petri dish and cured overnight at 60 °C to form a 5 mm-thick slab. The reservoir layer was then punched with 10 mm-diameter holes. The thin membrane was produced by spin-coating the same PDMS mixture at 1100 rpm for 15 s, resulting in a ~ 100 μm-thick film and cured under the same conditions. All layers were bonded sequentially after oxygen plasma treatment (PE-25, Plasma Etch Inc., USA) (Fig. 1a) and post-cured on a hot plate at 80 °C for 12 h to strengthen the bonds. Prior to cell culture, chips were sterilized in 80% ethanol and thoroughly rinsed with DIW.

### Control and characterization of the curvature array chip

The thin PDMS layer was inflated to form three distinct curvatures (low, medium, and high) by injecting DIW through the inlet port of the hydraulic chamber using a syringe (309659, BD, USA). The detailed procedure for generating and controlling curvature, based on the self-recovery properties of PDMS, is described in Supplementary Note 7. To measure the curvature angle, a horizontal baseline and a perpendicular axis were defined as references in the images. Three points—the apex of the dome, the intersects of the dome with the baseline, and the midpoint along this arc—were marked using the ROI manager in ImageJ (NIH, Bethesda, Maryland, USA). The resulting angle was computed in ImageJ using circular segment formulas and converted to diopters. Additional time-lapse volumetric characterizations of curvature under a flow rate of 60 μL/min are provided separately in Supplementary Note 8.

### Surface functionalization for cell culture

Surface modification of PDMS was necessary to support stable cell culture under curvature. We coated the surface with polydopamine (PDA), which covalently links collagen to the PDMS by forming a PDA layer via oxidative polymerization. PDA concentration was optimized (Supplementary Note 3) for uniform ECM coverage and robust cell

adhesion even at high- curvature. Briefly, PDMS surfaces were first plasma-treated for 1 min, then immersed in 0.01% (w/v) dopamine hydrochloride solution (H8502, Sigma-Aldrich, USA) dissolved in 1.0 M Tris-HCl buffer (pH 8.5, MB-027-1000, Rockland, USA) and incubated for 24 h. After rinsing twice with DIW and drying, the surfaces were coated with collagen I (100 µg/mL; 5005, PureCol®, Advanced BioMatrix, USA) overnight. Coated chips were rinsed, UV-sterilized for 1 h, and stored until use.

## Cell culture and droplet cell seeding

Primary human corneal keratocytes (HCK; #6520, ScienCell, USA) were cultured in Fibroblast Medium (#2301, ScienCell) supplemented with 2% fetal bovine serum (FBS; #0010), 1% fibroblast growth supplement (FCS; #2352), and 1% penicillin-streptomycin (P/S; #0503) on poly-L-lysine−coated flasks (10 mg/mL; #0413). Cells were expanded and used for experiments between passages 2−3. Cells were then resuspended at two concentrations ($2.5 \times 10^5$ cells/mL and $1 \times 10^6$ cells/mL) and loaded into 3 mL empty cartridges (CSC010300102, CELLINK Inc.). Droplet seeding was performed at the center of each well of the curvature array chip using a 3D bioprinter (Bio X™, CELLINK) fitted with a 25 G needle. Initial cell attachment was confirmed by staining with Hoechst 33258 (#H3569, Invitrogen) and cell coverage was quantified in ImageJ by counting nuclei and measuring the coverage area (Supplementary Note 4, 5, 9).

## Cell transformation: Corneal fibroblasts and myofibroblasts

Quiescent keratocytes, fibroblasts, and myofibroblasts were derived from primary human corneal keratocytes (HCK; ScienCell) by modulating culture media and subsequently cultured on the curvature chip. Quiescent keratocytes were maintained in serum-free DMEM/F12 with 1% penicillin/streptomycin (P/S), 1 mM L-ascorbic acid 2-phosphate (49752, Sigma-Aldrich), and 1% transferrin−selenium solution (I3146, Sigma-Aldrich). Fibroblasts were induced by culturing HCKs in DMEM/F12 containing 10% FBS, 1% P/S, and 10 ng/mL basic fibroblast growth factor (bFGF; 233-FB, R&D Systems). Myofibroblasts were generated by culturing HCKs in DMEM/F12 with 10% FBS and 10 ng/mL transforming growth factor-β (TGF-β; 240-B-002, R&D Systems). All cell types were expanded on flat substrates for three days, validated, and then seeded onto the chip. Medium was replaced every other day, and chips were kept in sterile Petri dishes in a humidified incubator at 37 °C, 5% CO$_2$.

## Immunostaining and image acquisition

Wells of the chips were rinsed with PBS, fixed in 10% formalin for 15 min, washed, and deflated to flatten the dome for fluorescence imaging. Samples were permeabilized with 0.1% Triton X-100 for 15 min at room temperature. For intracellular markers such as α-SMA and ALDH3A1, fixation and permeabilization were performed using a 1:1 acetone/methanol solution at −20 °C. After blocking with 1% BSA, samples were incubated with primary antibodies overnight at 4 °C, washed with PBS, and incubated with secondary antibodies for 1 h at room temperature. Nuclei were stained with Hoechst 33258. Primary and secondary antibodies are detailed in Supplementary Table 2. Images were acquired on a Nikon Eclipse Ti microscope with NIS-Elements software, stitched at 40× magnification, keeping exposure settings constant.

## Immunofluorescence image analysis

Quarter-circle images from each well were captured using stitched 40× magnification fields, then segmented into 1024 × 1024 pixel tiles for batch analysis using custom MATLAB R2022b code (MathWorks). For ALDH3A1, α-SMA, and vinculin images, mean green and red fluorescence intensities were computed after Gaussian filtering, thresholding, RGB channel separation, and background subtraction using a rolling-ball algorithm (radius 10−20 pixels). Correlated total cell fluorescence

(CTCF) was calculated as: CTCF = Integrated density − (Area × Mean background). CTCF was normalized to nucleus count and averaged over all tiles. Region-specific fluorescence was quantified relative to curvature-induced displacement boundaries (Supplementary Note 2).

## Cell alignment analysis

Cell orientation was quantified using the OrientationJ plugin (Biomedical Imaging Group, EPFL, Switzerland) in ImageJ[67], with the "gradient" method with cubic spline interpolation. F-actin images were preprocessed by adjusting brightness and contrast, followed by thresholding to produce binary images with dark backgrounds, which were exported as 32-bit TIFF files. Orientation angles from −90° to 90° were extracted, and the frequency of white pixels at each angle was calculated. To enable comparison across curvature conditions, frequency distributions were normalized by dividing each value by the maximum frequency, yielding a scaled range where the peak frequency equals one.

## RNA isolation and RT-PCR gene level quantification

Total RNA was pooled from at least three independent chips per group at day 9, extracted with the Direct-zol RNA Miniprep Kit (R2051, Zymo Research, USA), and reverse-transcribed from 0.2 µg of total RNA using the High-Capacity cDNA Reverse Transcription Kit (4368814, Applied Biosystems, USA). RT-qPCR was performed using Power SYBR Green Master Mix (4367659, Applied Biosystems, USA) and primers (Supplementary Table 3) targeting ECM markers (*COL1A1, COL3A1, COL5A1, lumican, and keratocan*) and phenotype markers (*ALDH3A1 and α-SMA*) with GAPDH as the housekeeping gene. Relative expression levels were calculated using the $2^{-\Delta\Delta Ct}$ method and normalized to the flat condition.

## RNA sequencing and Bioinformatic Analysis

Total RNA was extracted as above, quality-checked by NanoDrop spectrophotometer (Thermo Fisher Scientific, USA), and Novogene Inc. (Sacramento, CA, USA) using an Agilent 5400 Bioanalyzer (Agilent Technologies, USA). RNA sequencing was conducted on the Illumina NovaSeq X Plus platform, generating 40−60 million 150 bp paired-end reads per sample with poly-A mRNA enrichment. Libraries were prepared using the NEBNext Ultra II RNA Library Prep Kit (New England BioLabs, USA). Quality metrics showed >95% of bases with Q30 scores and a base-calling error rate of 0.01%, with an average GC content of ~51%. For bioinformatic analysis, raw reads were assessed with FastQC and trimmed using Trimmomatic. Clean reads were aligned to the GRCh38/hg38 reference genome using STAR, and gene expression was quantified with featureCounts. Differential expression analysis was performed with DESeq2, applying an adjusted *p*-value threshold of <0.05. Functional enrichment analysis, including Gene Ontology (GO) and Kyoto Encyclopedia of Genes and Genomes (KEGG) pathway analysis, was performed using ClusterProfiler in R.

## Statistical analysis

Experiments were conducted across three curvature levels (high, medium, and low) and a flat control. Data were obtained from three independent biological replicates (N = 3), and results are presented as mean ± standard deviation (s.d.), with the number of analyzed samples indicated in the figure legends (n > 5). Immunostaining was quantified from hundreds of images (n values shown in graphs). After seeding all chips under identical flat conditions, samples were randomly allocated in equal numbers to flat, low, medium, and high curvature groups, and the assigned curvature was subsequently applied. To minimize variability, all experiments used cells of the same passage number, identical seeding densities, and uniform coating and media conditions. No statistical methods were used to predetermine sample size. Sample sizes were chosen based on previous studies using comparable organ-on-chip and in vitro corneal stromal cell models, which established

that N = 3 independent biological replicates per condition are sufficient to capture reproducible trends. For image-based quantifications, more than 50 cells or images were analyzed per condition, providing adequate statistical power to detect differences across curvature groups. Immunostaining was quantified from hundreds of images (n values shown in graphs). Box plots display the median (center line), 25th and 75th percentiles (box limits), and whiskers extending to 1.5× the interquartile range (IQR); mean values are denoted by ◆. Fluorescence intensity was analyzed across global and spatially defined regions (center, slope, and edge) of each well. Statistical significance was assessed by one-way ANOVA followed by Scheffé's post hoc test ($P < 0.05$, $P < 0.01$, $P < 0.001$; see Figs. 2–4, and 6, Supplementary Figs. 8 and 10, and Supplementary Fig. 13). Graphical and statistical analyses were performed using Excel (Microsoft), OriginPro 2023b (OriginLab), SPSS 29.0 (IBM), and MATLAB.

### Reporting summary
Further information on research design is available in the Nature Portfolio Reporting Summary linked to this article.

## Data availability
Source data underlying the graphs presented in the main figures and supplementary data and figures are provided with this paper. All other raw and processed datasets generated during this study are available from the corresponding author upon request and will be provided within four weeks. RNA-sequencing data have been deposited in the Gene Expression Omnibus (GEO) under accession code GSE308374. Source data are provided with this paper.

## Code availability
The MATLAB code for quantifying fluorescent intensity is available in the Supplementary Information (Supplementary Software 1 section). For further inquiries, please contact the corresponding author.

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

## Acknowledgements

This work was supported by an Ascender Grant from the University of Utah Research Foundation, NIH grants UL1TR002538, R01EY022076, P30EY014800, and an Unrestricted Grant from Research to Prevent Blindness to the Department of Ophthalmology and Visual Sciences at the University of Utah.

## Author contributions

J.K. and M.K conceived of the overall research goals and aims; J.K. and M.K. were responsible for the overall engineering design of the organ chip; M.K. was involved in the manufacturing of organ chip; M.K. performed for the biological experiments; M.K., K.C., D.K., and J.K. were responsible for the biological analysis of fluorescent images and gene expressions; K.C., M.K. and J.K. were responsible for simulating curvature data; M.K. and J.K. were responsible for original drafting of the manuscript; All authors contributed to the reviewing and editing of the manuscript.

## Competing interests

M.K., K.C., and J.K. are inventors on a patent application (U.S. Patent Application No. 18/196,932) filed with the United States Patent and Trademark Office, related to the technology described in this work. The application is currently under review. D.K. declares no competing interests.
