## [Transparent Peer Review file · Nature Communications]

Regulation of corneal stromal cell behavior by modulating curvature using a hydraulically-controlled organ chip array

Corresponding Author: Professor Jungkyu Kim

Version 0:

Reviewer comments:

Reviewer #1

(Remarks to the Author)

Originality and significance: The authors present an interesting study of the behavior of corneal stromal cells on surfaces with variable curvature using an original hydraulically controlled organ chip array. This system is well described and could be very useful to the research community interested in the effects of curvature on cells. Overall, the work is of interest to this last community but also probably to the community concerned by the cornea pathologies associated with corneal shape dysfunctions.

Indeed, the authors compare the behavior of 3 important corneal cell types: corneal keratocytes, fibroblasts and myofibroblasts on 4 different levels of curvatures (flat, low, medium and high). Those three different curvatures are determined using the circular segment formulas based on the characteristics of the anterior surface of the cornea and this is a strength and originality of this work.

The biological analyses are mainly based on immunostaining of intracellular proteins allowing to exhibit phenotypic alterations in response to curvature. With this objective, cell morphology, cell alignment cytoskeleton and focal adhesions organization as well as contractile and extracellular matrix proteins expression are analyzed. Immunostaining observations are sometimes confirmed by gene expression analysis.

Key results: The authors show that keratocytes exhibit phenotypic alterations in response to curvature changes, notably including a decrease in ALDH3 expression and an increase in α -SMA expression. For focal adhesion, corneal fibroblast and myofibroblasts showed enhanced vinculin localization in response to curvature, while corneal keratocytes presented reduced vinculin expression. For cell alignment and ECM expression, most stromal cells showed a radially organized f-actin and collagen fibrils under all curvatures. Interestingly, for corneal fibroblast under medium curvature, an orthogonal cell alignment was observed. Furthermore, lumican expression was upregulated in corneal keratocytes, and keratocan expression was increased in corneal fibroblasts and myofibroblasts due to curvature.

Data & methodology: Some weaknesses in the interpretation of certain results was observed, particularly those obtained using immunolabelling. Indeed, the majority of the work is based on the quantification of red and green fluorescence intensities after immunolabeling thanks to a custom MATLAB code. This method is adequate for homogeneous labeling but is less pertinent to quantify localized proteins such as vinculin. In this case, contrary to what is expected, it appears on Figure 4 and Supplementary Figure 6 that the vinculin immunolabeling is rather diffuse all over the cells (fibroblasts and myofibroblasts) and looks like an actin labeling. The same can be observed on Supplementary Figure 3b. Thus, the specificity of this labeling appears too weak. Consequently, it is likely that the interpretation drawn from the quantification of this immunolabeling is not sufficiently precise and may even be erroneous. The authors should revise their immunolabeling or improve their images to guarantee its specificity.

Similarly, it is surprising to see nuclei labelled with anti-collagen type I antibody in fibroblasts on Supplementary Figure 12. Logically, collagen should be only labelled when synthesized in the cytoplasm. The authors should revise their immunolabeling or improve their images. This could be also completed by a collagen type I gene expression study at day 6 to complete results at Day 9 given on Figure 6.

Secondly, the authors describe an analysis of cell viability in Supplementary Figure 2 and Supplementary section 1 but they don't explain anywhere the protocol they followed. The only information given in Supplementary Figure 2 caption is a reading of absorbance at 450nm. This information should be given.

Suggested minor improvements:

- Figure 4 and Supplementary Figure 6: the magnification seems different between these 2 figures while the bars are the same. Please check.
- It's not clear for the reviewer why flat surfaces are abbreviated by C (instead of F) while the other low, medium and high curvatures are respectively abbreviated by L, M and H in Figures 2, 3, 5 and 6.
- Supplementary section 2: change Supplementary Figure 4 for Supplementary Figure 3.
- Supplementary section 3: change Supplementary Figure 6 for Supplementary Figure 7, Supplementary Figure 7 for Supplementary Figure 8, Supplementary Figure 8 for Supplementary Figure 9.
- Supplementary section 6: Change Supplementary Fig. 15a for Supplementary Fig. 15c.
- Chapter 4. Methods:
 - § 4.6: remove the list of antibodies after Supplementary Table 2 since they are listed in this Table.
 - § 4.9: change Supplementary Table 1 for Supplementary Table 3
 - § 4.10: Change first sentence for: Experimental trials were conducted across three distinct curvature levels (high, medium, and low) in addition to a curvature-free flat group for comprehensive comparative analysis.
- References: the format of references is not homogenous. Several times, 3 or 4 redundant references are given in the discussion. The authors should choose the most pertinent ones.

In addition, the manuscript is difficult to read given the large number of references to additional material or figures throughout. An effort should be done by the authors to reduce the number of references to additional material.

Reviewer #2

(Remarks to the Author)

To investigate the impact of curvature stress on corneal keratocytes, fibroblasts, and myofibroblasts, the authors employed a hydraulically controlled curvature array chip capable of adjusting curvature levels to mimic both healthy and diseased corneal conditions. Their findings at the organ-on-a-chip level revealed that curvature significantly influences the phenotype of corneal stromal cells and the structural organization of corneal stromal tissue. This offers a promising avenue for addressing curvature-dependent corneal shape disorders. Nevertheless, there are several concerns that require further clarification and discussion from the authors before considering the manuscript for publication.

1. What does "n" represent in Figure 3b-c and Figure 4b?
2. Could you clarify the significance of the orthogonal pattern observed in cornea fibroblasts compared to previous studies that used different methods?
3. The author elaborates on the influence of curvature on corneal stromal cell behavior. Is there any hope in the future to restore corneal dysfunction or promote wound healing by adjusting curvature?
4. Will the model be used next to explore molecular mechanisms related to ECM deposition and degradation to reveal the effects of corneal shape deformation?

Reviewer #3

(Remarks to the Author)

I commend the authors for embarking on an important exploratory study determining the impact of corneal curvature on keratocyte, fibroblast and myofibroblast phenotype. There is indeed a critical knowledge gap in the impact of curvature on corneal cell behavior and phenotype as acknowledged in a recent review of corneal mechanobiology (<https://doi.org/10.1016/j.preteyeres.2023.101234>). The microfabricated curvature array chip is cleverly designed and is an important advance in this field. The relevance in particular to the keratoconus field is particularly high and I encourage the authors to acquire primary cells from keratoconic patients to determine how they would respond to curvature changes in future studies.

MAJOR CONCERNS:

The sole focus on cell phenotype with no exploration of underlying mechanisms is a key limitation of the study. While RhoA/ROCK pathways are conjectured to be modulating these phenotypic changes, little data is provided in support. For example, the RhoA/ROCK pathway could be inhibited to determine if the phenotype changes are modulated. Much of the discussion is conjecture with TGF- β /SMAD signaling postulated to be involved but no data are provided. The vinculin data just showed trends so the entire discussion of those changes is overinterpretation of that data as detailed below. The discussion is also quite long given the amount of data presented and should be shortened to increase impact and clarity.

2.4 All the vinculin data were trends correct? Given that sample size was not an issue, is it appropriate to say that vinculin was altered by curvature? Your data suggests otherwise and it is an overinterpretation to say that it was different.

Figure 4: No asterisks indicate statistical significance between groups in the bar graphs. Were there significant differences between curvatures? If so, please denote. If not, I suggest removing the figure as it does not demonstrate curvature-influenced focal adhesion differences.

MINOR CONCERNS:

2.3 Were the protein and mRNA concentrations for α SMA and ALDH3A1 consistent or inconsistent with each other? It would be easier to interpret the data if they are presented together rather than separate figures and paragraphs.

Figures 2, 3, 4 and 6: The C, L, M and H are undefined in the figure legends. Given the large number of cells analyzed, violin or box plots would better represent the spread of the data (consider for Suppl Figure 3 as well).

Figures 3 and 4: Labeling is inconsistent. Flat is used for the IHC but bar graphs use C (presumably for control).

Figure 5: These results are interesting but only subjective differences are shown. Can these changes be quantified and statistically compared?

Supplementary Figure 9: This figure would be helpful to include as part of Figure 5 so the data is easier to interpret.

Supplementary Figure 10: Consider moving this figure to the main manuscript since the RhoA expression is essential to the discussion.

Supplementary Figure 12: Please denote significant differences in collagen expression. This figure should be moved to the main document given that the results are a feature of the discussion.

It is suggested that line numbers be included as it is much more difficult and time-consuming for the reviewer to denote precise locations where changes need to be made.

Minor grammatical errors should be addressed (eg. cornea stroma/mechanobiology should be corneal stroma/mechanobiology – this occurs numerous times in manuscript) and in 2.3 "the only high curvature" should be "only the high curvature). There are others and the manuscript should be carefully reviewed to facilitate readability.

Reviewer #4

(Remarks to the Author)

The manuscript by Kim et al. presents an interesting study that underscores the crucial role of curvature in cornea mechanobiology, specifically regarding phenotypic alterations and extracellular matrix remodeling. The use of hydraulically actuated PDMS to investigate curvature effects is innovative and noteworthy. The paper is generally well-written, providing ample detail and comprehensive results focused on its primary objective. However, the manuscript lacks an in-depth examination of the curvature chip itself, aside from presenting a series of images that illustrate the curvature angle over a 360-second duration. It is suggested that the authors expand the manuscript to include a more thorough mechanical and system description of the chip in a dedicated section. Also, incorporating images of the curvature at the conclusion of Day 3 may also help validate the chip's curvature stability and accuracy. Adding more discussion on how the findings could inform future cornea dystrophy treatment strategies is also recommended, as the current discussion leans heavily towards factual and direct impacts rather than potential applications. Other comments can be found below:

1. The motivation behind choosing PDMS-based curvature chips over other materials, such as polystyrene, which could potentially offer greater resistance to external vibrations or forces, is not clearly articulated. A detailed rationale for this choice would be beneficial.
2. A more detailed description of the cell culture conditions is needed. For instance, it's mentioned that cells are cultured for three days, but it's unclear whether the medium in the chip's reservoir was changed during this period, if the reservoir is open, the specific conditions within the incubator, the temperature of the DIW, and whether there is a decontamination protocol for the culture environment.
3. The discussion section should place more emphasis on the curvature's impact on cell behavior rather than the coating material.
4. The absence of a control group in the cell viability data is notable.
5. The rationale for selecting the 3, 6, and 9-day marks for the study is not explained.
6. The acronym DIW should be fully spelled out upon its first occurrence.
7. There are grammatical errors throughout the manuscript. A thorough review and revision of the text are recommended to enhance its clarity and professionalism.

Version 1:

Reviewer comments:

Reviewer #1

(Remarks to the Author)

I read with attention the revision of the manuscript NCOMMS-24-10590 from Kim, M. et al. "Regulation of Corneal Stromal Cell Behavior by Modulating Curvature using a Hydraulically Controlled Organ Chip Array".

The manuscript was very well improved following the reviewer's comments. In particular, the addition of dot plots for quantification of immunostaining data on Figure 3 and Figure 4 improved the demonstration of authors assertions. Moreover, the addition of RNA sequencing completed very well the qRT-PCR to examine curvature-induced transcriptomic changes across the different tested cell types. These whole-genome sequencing analyses enabled the authors to go much further in their discussion of the biological processes involved in the response of the three cell types tested to curvature. In particular, the authors were able to determine the signaling pathways involved and discuss more deeply the differences between fibroblasts, myofibroblasts, and keratocytes.

In conclusion, I consider that the manuscript has been considerably improved and can now be published.

However, I suggest a few minor corrections be made prior to publication:

p8: I do not understand the angles of $99.99^\circ \pm 3.73^\circ$ given by the authors in Figure 5a, whereas in Figure 5a no angles are given. This is surely Figure 5b. However, only angles between -90° and $+90^\circ$ are given on figure 5b... This point will need to be reviewed.

p8: the authors refer to a supplementary figure, Fic.14c, which does not exist.

Reviewer #2

(Remarks to the Author)

This study presents a hydraulically controlled curvature array chip to investigate how geometric cues regulate corneal stromal cell behavior. The authors characterized curvature-induced changes in proliferation, phenotype (ALDH3A1/ α -SMA), focal adhesion (vinculin/pFAK), alignment, and ECM remodeling in keratocytes, fibroblasts, and myofibroblasts. The work addresses a critical gap in corneal mechanobiology, with implications for ectatic diseases (e.g., keratoconus). The revised manuscript has addressed most of the reviewers' concerns through additional data and improved presentation. However, I have few remaining concerns and suggestions:

1. While RNA-seq identifies RhoA/ROCK and TGF- β as candidate pathways, functional validation (e.g., pharmacological inhibition) is absent. The rebuttal appropriately notes this as a future direction, but the discussion should more explicitly acknowledge this limitation.
2. The expanded discussion on curvature modulation for wound healing/disease management (p. 21) is appreciated but remains speculative. A brief mention of specific testable strategies (e.g., curvature-adjusting implants) would enhance the impact.
3. The significance of fibroblast orthogonal organization (Fig. 5) is now contextualized against prior studies using chemical/patterned substrates. However, quantitation of alignment efficiency (e.g., % cells aligned within $\pm 10^\circ$ of orthogonal axes) would strengthen claims.

Minor revisions:

1. Terminology and grammar require refinement to ensure professionalism. For example, "DIW" should be spelled out as "deionized water" at first use. In the Abstract, "were sufficient" should be "was sufficient". In the Discussion, "e.g., "tenascin-C (TNC), a matrix glycoprotein [is] involved". A thorough language edit is recommended.
2. Methodological transparency can be improved. Figure legends quantifying data (e.g., Fig. 3b-c) should clarify "n" (e.g., "n = number of images/cells analyzed"). Critical technical details—such as the use of a 30G syringe needle and 60 μ L/min flow rate for hydraulic actuation—currently relegated to Supplementary Section 7, should be moved to the main Methods to enhance reproducibility.

In general, the authors have comprehensively addressed reviewer concerns through additional experiments, data re-analysis, and textual revisions. I recommend acceptance with minor revisions, after the authors further refine the text according to above comments.

Reviewer #3

(Remarks to the Author)

The authors have done an exceptional job responding to concerns raised by the reviewers and provide critical additional data in support of their original observations. In particular, the addition of the RNAseq data combined with quantification of IHC expression are strengths of the revised manuscript. The methods are highly novel and experiments conducted with rigor. This work greatly advances our understanding of how corneal curvature impacts stromal cell phenotype and gene expression.

MINOR CONCERNS:

Intro, Suggest changing "corneal haziness" to "corneal haze"

Intro, Paragraph 3 This is an oversimplification of key growth factors in keratocyte-fibroblast-myofibroblast transformation.

TGF-b also can drive keratocyte to fibroblast differentiation. I suggest reviewing this publication which nicely summarizes key growth factors in corneal wound healing within the context of different ECM environments. 10.1038/s41598-023-37776-9 Methods, Cell Culture... What passage number of cells were used?

Figure 3 and Supplementary Figure 10 DAPI is used to mark nuclei (blue) in the merged images but not denoted in the figure legend.

Reviewer #4

(Remarks to the Author)

The authors have addressed my previous comments and provided sufficient details in the supplementary materials.

Detailed point-by-point response

Reviewer #1 (Remarks to the Author):

Originality and significance: The authors present an interesting study of the behavior of corneal stromal cells on surfaces with variable curvature using an original hydraulically controlled organ chip array. This system is well described and could be very useful to the research community interested in the effects of curvature on cells. Overall, the work is of interest to this last community but also probably to the community concerned by the cornea pathologies associated with corneal shape dysfunctions.

Indeed, the authors compare the behavior of 3 important corneal cell types: corneal keratocytes, fibroblasts and myofibroblasts on 4 different levels of curvatures (flat, low, medium and high). Those three different curvatures are determined using the circular segment formulas based on the characteristics of the anterior surface of the cornea and this is a strength and originality of this work.

The biological analyses are mainly based on immunostaining of intracellular proteins allowing to exhibit phenotypic alterations in response to curvature. With this objective, cell morphology, cell alignment cytoskeleton and focal adhesions organization as well as contractile and extracellular matrix proteins expression are analyzed.

Immunostaining observations are sometimes confirmed by gene expression analysis.

Key results: The authors show that keratocytes exhibit phenotypic alterations in response to curvature changes, notably including a decrease in ALDH3 expression and an increase in α -SMA expression. For focal adhesion, corneal fibroblast and myofibroblasts showed enhanced vinculin localization in response to curvature, while corneal keratocytes presented reduced vinculin expression. For cell alignment and ECM expression, most stromal cells showed a radially organized f-actin and collagen fibrils under all curvatures. Interestingly, for corneal fibroblast under medium curvature, an orthogonal cell alignment was observed. Furthermore, lumican expression was upregulated in corneal keratocytes, and keratocan expression was increased in corneal fibroblasts and myofibroblasts due to curvature.

Data & methodology: Some weaknesses in the interpretation of certain results was observed, particularly those obtained using immunolabelling. Indeed, the majority of the work is based on the quantification of red and green fluorescence intensities after immunolabeling thanks to a custom MATLAB code. This method is adequate for homogeneous labeling but is less pertinent to quantify localized proteins such as vinculin. In this case, contrary to what is expected, it appears on Figure 4 and Supplementary Figure 6 that the vinculin immunolabeling is rather diffuse all over the cells (fibroblasts and myofibroblasts) and looks like an actin labeling. The same can be observed on Supplementary Figure 3b. Thus, the specificity of this labeling appears too

weak. 1) Consequently, it is likely that the interpretation drawn from the quantification of this immunolabeling is not sufficiently precise and may even be erroneous. The authors should revise their immunolabeling or improve their images to guarantee its specificity.

RESPONSE: We thank the reviewer for the comment. To ensure reliability, all immunostaining data were quantified from multiple tiled images covering a quarter-sized region (~8 mm diameter) per sample, rather than from a single field of view. Quantification was performed on hundreds of cells per condition and presented as dot plots (Figure 3, Figure 4, Supplementary Figures 10 and 13). While vinculin labeling appeared diffuse in some cases, this distribution is a well-documented physiological pattern observed in both quiescent keratocytes and activated stromal cells such as fibroblasts, particularly under low adhesion or mechanical stress conditions, and does not indicate nonspecific staining. To further support our interpretation of focal adhesion remodeling, we included pFAK immunostaining (Supplementary Figure 12), which exhibited consistent curvature-dependent trends. Additionally, to eliminate the possibility of nonspecific binding, we repeated vinculin staining using a directly conjugated primary antibody. This approach yielded comparable results and further confirmed labeling specificity (Supplementary Figure 12, Supplementary Table 2).

2) Similarly, it is surprising to see nuclei labelled with anti-collagen type I antibody in fibroblasts on Supplementary Figure 12. Logically, collagen should be only labelled when synthesized in the cytoplasm. The authors should revise their immunolabeling or improve their images. This could be also completed by a collagen type I gene expression study at day 6 to complete results at Day 9 given on Figure 6.

RESPONSE: Upon careful consideration, we have decided to remove the collagen type I immunostaining image, as we agree that the signal appearance may be confusing. Regarding the suggestion to assess gene expression at Day 6, we would like to clarify that this time point represents only three days after curvature exposure, which is likely insufficient for robust collagen synthesis and matrix deposition. Since collagen remodeling is generally a delayed response, we reasoned that longer culture durations would be more appropriate for evaluating curvature-induced ECM production. Therefore, we performed RNA sequencing at Day 9 to comprehensively assess the transcriptional regulation of collagen-related and fibrosis-associated genes (Figure 7, Supplementary Figure 15, Supplementary Figure 16). These data provide deeper insight into ECM gene expression dynamics in response to curvature and complement our protein-level analysis conducted at later time points.

3) Secondly, the authors describe an analysis of cell viability in Supplementary Figure 2 and Supplementary section 1 but they don't explain anywhere the protocol they

followed. The only information given in Supplementary Figure 2 caption is a reading of absorbance at 450nm. This information should be given.

RESPONSE: The detailed protocol for the cell viability assay is provided in Supplementary Section 3. This includes information on the specific assay kit used, incubation times, and measurement procedures.

Suggested minor improvements:

-Figure 4 and Supplementary Figure 6: the magnification seems different between these 2 figures while the bars are the same. Please check.

-It's not clear for the reviewer why flat surfaces are abbreviated by C (instead of F) while the other low, medium and high curvatures are respectively abbreviated by L, M and H in Figures 2, 3, 5 and 6.

-Supplementary section 2: change Supplementary Figure 4 for Supplementary Figure 3.

-Supplementary section 3: change Supplementary Figure 6 for Supplementary Figure 7, Supplementary Figure 7 for Supplementary Figure 8, Supplementary Figure 8 for Supplementary Figure 9.

-Supplementary section 6: Change Supplementary Fig. 15a for Supplementary Fig. 15c.

-Chapter 4. Methods:

§ 4.6: remove the list of antibodies after Supplementary Table 2 since they are listed in this Table.

§ 4.9: change Supplementary Table 1 for Supplementary Table 3

§ 4.10: Change first sentence for: Experimental trials were conducted across three distinct curvature levels (high, medium, and low) in addition to a curvature-free flat group for comprehensive comparative analysis.

-References: the format of references is not homogenous. Several times, 3 or 4 redundant references are given in the discussion. The authors should choose the most pertinent ones.

In addition, the manuscript is difficult to read given the large number of references to additional material or figures throughout. An effort should be done by the authors to reduce the number of references to additional material.

RESPONSE: We thank the reviewer for these suggestions. All minor comments have been addressed. To enhance overall readability, we revised the manuscript to emphasize key findings and reduced excessive references to supplementary materials.

Reviewer #2 (Remarks to the Author):

To investigate the impact of curvature stress on corneal keratocytes, fibroblasts, and myofibroblasts, the authors employed a hydraulically controlled curvature array chip capable of adjusting curvature levels to mimic both healthy and diseased corneal conditions. Their findings at the organ-on-a-chip level revealed that curvature significantly influences the phenotype of corneal stromal cells and the structural organization of corneal stromal tissue. This offers a promising avenue for addressing curvature-dependent corneal shape disorders. Nevertheless, there are several concerns that require further clarification and discussion from the authors before considering the manuscript for publication.

1. What does "n" represent in Figure 3b-c and Figure 4b?

RESPONSE: The number of images analyzed is indicated as 'n,' as described in Section 5.11, Statistical Analysis.

2. Could you clarify the significance of the orthogonal pattern observed in cornea fibroblasts compared to previous studies that used different methods?

RESPONSE: We thank the reviewer for the comment. As requested, we have clarified the significance of the orthogonal alignment pattern by briefly comparing it to conventional uniaxial and biaxial stretch models (4th paragraph in the Introduction), and we elaborated further on this comparison in lines 3–4 of the discussion section. Additionally, we discussed how our mechanically induced alignment differs from previous studies that relied on chemical agents or patterned substrates to achieve similar orthogonal organization (see “Cell alignment” in the Discussion section). These revisions highlight how curvature serves as a physiologically relevant mechanical cue that uniquely directs stromal cell organization.

3. The author elaborates on the influence of curvature on corneal stromal cell behavior. Is there any hope in the future to restore corneal dysfunction or promote wound healing by adjusting curvature?

RESPONSE: We appreciate the reviewer’s question. In our study, we observed that increasing curvature promotes fibrotic phenotypes in corneal stromal cells, along with changes in ECM-related gene expression. This suggests that curvature acts not only as a passive structural feature but also as an active regulator of cell behavior. We agree that this finding holds promise for future therapeutic strategies to restore ECM homeostasis or promote wound healing in ectatic conditions, where curvature and ECM remodeling are both disrupted. To address this, we have added a statement at the end of the Discussion

emphasizing the need for further investigations into curvature-driven MMP/TIMP regulation and collagen synthesis. Long-term culture models will be essential to fully explore the regenerative potential of curvature modulation.

4. Will the model be used next to explore molecular mechanisms related to ECM deposition and degradation to reveal the effects of corneal shape deformation?

RESPONSE: We thank the reviewer for this valuable question. Indeed, one of the key motivations behind developing our curvature-based model was to enable mechanistic studies of ECM remodeling in response to corneal curvature changes. To support this goal, we performed RNA sequencing (Figure 7, Supplementary Figure 15, Supplementary Figure 16) to examine curvature-induced transcriptomic changes across different stromal cell types. These data revealed the patterns of ECM remodeling and identified cell type-specific differences in fibrosis-associated gene expression.

Reviewer #3 (Remarks to the Author):

I commend the authors for embarking on an important exploratory study determining the impact of corneal curvature on keratocyte, fibroblast and myofibroblast phenotype. There is indeed a critical knowledge gap in the impact of curvature on corneal cell behavior and phenotype as acknowledged in a recent review of corneal mechanobiology (<https://doi.org/10.1016/j.preteyeres.2023.101234>). The microfabricated curvature array chip is cleverly designed and is an important advance in this field. The relevance in particular to the keratoconus field is particularly high and I encourage the authors to acquire primary cells from keratoconic patients to determine how they would respond to curvature changes in future studies.

MAJOR CONCERNS:

1) The sole focus on cell phenotype with no exploration of underlying mechanisms is a key limitation of the study. While RhoA/ROCK pathways are conjectured to be modulating these phenotypic changes, little data is provided in support. For example, the RhoA/ROCK pathway could be inhibited to determine if the phenotype changes are modulated. Much of the discussion is conjecture with TGF- β /SMAD signaling postulated to be involved but no data are provided. The vinculin data just showed trends so the entire discussion of those changes is overinterpretation of that data as detailed below. The discussion is also quite long given the amount of data presented and should be shortened to increase impact and clarity.

RESPONSE: We appreciate the reviewer's comment. Our study aimed to characterize curvature-induced fibrotic responses, including ECM remodeling and phenotype transitions, across three distinct corneal stromal cell types. To examine broad mechanobiological responses, we performed RNA sequencing, which provided clear, cell-type-specific transcriptional profiles and revealed broad pathway involvement in response to curvature. However, we agree that while RNA sequencing data illuminate the breadth of curvature-responsive pathways, pharmacological inhibition or genetic knockout approaches would provide more straightforward answers regarding the functional roles of specific pathways such as RhoA/ROCK and TGF- β /Smad. This limitation has been acknowledged and incorporated as a future direction in the final paragraph of the Discussion, with references. Additionally, we have revised the manuscript to improve clarity and emphasize the central findings.

2) 2.4 All the vinculin data were trends correct? Given that sample size was not an issue, is it appropriate to say that vinculin was altered by curvature? Your data suggests otherwise and it is an overinterpretation to say that it was different.

RESPONSE: We appreciate this constructive feedback. As shown in Supplementary Figure 13, overall vinculin expression exhibited curvature-dependent differences. Notably, keratocytes and fibroblasts demonstrated a

decreasing trend in vinculin intensity with increasing curvature at D9. These observations were supported by additional experiments using pFAK immunostaining and vinculin immunofluorescence (Supplementary Figure 12), both showing consistent patterns. While statistically significant differences were present in the vinculin data, they were not explicitly marked in the original figure presentation. In the revised version, we have presented the data using dot plots with clear statistical significance indicators to improve data transparency and facilitate interpretation.

3) Figure 4: No asterisks indicate statistical significance between groups in the bar graphs. Were there significant differences between curvatures? If so, please denote. If not, I suggest removing the figure as it does not demonstrate curvature-influenced focal adhesion differences.

RESPONSE: We thank the reviewer for the helpful comment. In response, we have revised Figure 4 by presenting the focal adhesion data as dot plots instead of bar graphs, enabling clearer visualization of individual data points and statistical differences across curvature conditions. Statistical significance is now indicated, with further details provided in Section 5.11 (Statistical Analysis).

MINOR CONCERNS:

2.3 Were the protein and mRNA concentrations for aSMA and ALDH3A1 consistent or inconsistent with each other? It would be easier to interpret the data if they are presented together rather than separate figures and paragraphs.

Figures 2, 3, 4 and 6: The C, L, M and H are undefined in the figure legends. Given the large number of cells analyzed, violin or box plots would better represent the spread of the data (consider for Suppl Figure 3 as well).

Figures 3 and 4: Labeling is inconsistent. Flat is used for the IHC but bar graphs use C (presumably for control).

Figure 5: These results are interesting but only subjective differences are shown. Can these changes be quantified and statistically compared?

Supplementary Figure 9: This figure would be helpful to include as part of Figure 5 so the data is easier to interpret.

Supplementary Figure 10: Consider moving this figure to the main manuscript since the RhoA expression is essential to the discussion.

Supplementary Figure 12: Please denote significant differences in collagen expression. This figure should be moved to the main document given that the results are a feature of the discussion.

It is suggested that line numbers be included as it is much more difficult and time-consuming for the reviewer to denote precise locations where changes need to be made.

Minor grammatical errors should be addressed (eg. cornea stroma/mechanobiology should be corneal stroma/mechanobiology – this occurs numerous times in manuscript)

and in 2.3 "the only high curvature" should be "only the high curvature). There are others and the manuscript should be carefully reviewed to facilitate readability.

RESPONSE: We thank the reviewer for the helpful suggestions. All minor revisions have been implemented accordingly. To enhance overall readability, we revised the manuscript to emphasize key findings and to minimize references to supplementary materials where appropriate.

Reviewer #4 (Remarks to the Author):

The manuscript by Kim et al. presents an interesting study that underscores the crucial role of curvature in cornea mechanobiology, specifically regarding phenotypic alterations and extracellular matrix remodeling. The use of hydraulically actuated PDMS to investigate curvature effects is innovative and noteworthy. The paper is generally well-written, providing ample detail and comprehensive results focused on its primary objective.

1) However, the manuscript lacks an in-depth examination of the curvature chip itself, aside from presenting a series of images that illustrate the curvature angle over a 360-second duration. It is suggested that the authors expand the manuscript to include a more thorough mechanical and system description of the chip in a dedicated section. Also, incorporating images of the curvature at the conclusion of Day 3 may also help validate the chip's curvature stability and accuracy.

RESPONSE: We thank the reviewer for this constructive suggestion. In response, we have included a long-term stability test conducted over 9 days, which is now presented in Supplementary Figure 3 and described in Section 2.1, "Design and characterization of a curvature array chip." The stability test results demonstrate that curvature was consistently maintained throughout the cell culture experiments.

2) Adding more discussion on how the findings could inform future cornea dystrophy treatment strategies is also recommended, as the current discussion leans heavily towards factual and direct impacts rather than potential applications.

RESPONSE: We performed RNA sequencing across all curvature conditions for each stromal cell type (Figure 7, Supplementary Figures 15 and 16) to uncover curvature-induced transcriptional responses relevant to corneal ectatic dystrophies. These data revealed ECM remodeling and fibrotic signaling pathways associated with abnormal curvature, supporting the notion that altered corneal geometry may actively contribute to disease pathogenesis (as discussed in the second-to-last paragraph of the Discussion). Based on these findings, we expanded the Discussion to emphasize the need for long-term studies investigating MMP/TIMP regulation and collagen synthesis. Such studies will be essential to evaluate the therapeutic potential of curvature modulation in restoring stromal homeostasis and promoting corneal repair.

3) Other comments can be found below:

3.1. The motivation behind choosing PDMS-based curvature chips over other materials, such as polystyrene, which could potentially offer greater resistance to external vibrations or forces, is not clearly articulated. A detailed rationale for this choice would be beneficial.

RESPONSE: We selected PDMS as the chip material due to its elasticity, which is essential for generating controlled curvature. Unlike rigid materials such as polystyrene, PDMS allows reversible deformation and self-sealing after water injection using a 30G needle and syringe (described in Supplementary Section 7). This elastic behavior eliminates the need for continuous external actuation. Importantly, the elastic modulus of the PDMS membrane, as shown in Supplementary Figure 1, lies within the typical Young's modulus range of human cornea. The chip design and rationale are detailed in Section 2.1 (Design and characterization of a curvature array chip) and illustrated in Supplementary Figure 19.

3.2. A more detailed description of the cell culture conditions is needed. For instance, it's mentioned that cells are cultured for three days, but it's unclear whether the medium in the chip's reservoir was changed during this period, if the reservoir is open, the specific conditions within the incubator, the temperature of the DIW, and whether there is a decontamination protocol for the culture environment.

RESPONSE: We appreciate the reviewer's comment regarding the need for a more detailed description of the cell culture conditions. In response, we have added the relevant culture details to Section 5.5 (Cell differentiation: Corneal fibroblasts and myofibroblasts) and Supplementary Section 7.

3.3. The discussion section should place more emphasis on the curvature's impact on cell behavior rather than the coating material.

RESPONSE: In response to the reviewer's suggestion, we have relocated discussions on coating effects to Supplementary Section 3 and revised the main Discussion to focus on the impact of curvature on stromal cell behavior.

3.4. The absence of a control group in the cell viability data is notable.

RESPONSE: Cells failed to adhere stably to physisorption-based ECM coatings on curved PDMS surfaces, preventing cellular behavior studies. To address this issue, we introduced dopamine-assisted coating to enhance stability. The cell viability investigation aimed to optimize dopamine concentration for biocompatibility and cell viability under curved conditions, rather than evaluate dopamine's general efficacy. Therefore, physisorption-based ECM coating was not included as a control, since poor adhesion made it unsuitable for meaningful viability comparisons. Detailed explanation is provided in Supplementary Section 3.

3.5. The rationale for selecting the 3, 6, and 9-day marks for the study is not explained.

RESPONSE: We appreciate the reviewer's comment regarding the rationale for selecting the 3-, 6-, and 9-day time points. These time points were strategically chosen to capture the temporal dynamics of curvature-induced mechanobiological responses in corneal stromal cells. Day 3 captured initial cellular responses following curvature application and allowed assessment of early attachment patterns. Day 6 was selected to examine intermediate mechanotransduction responses, including focal adhesion remodeling and α -SMA expression changes. Day 9 served to evaluate mature cellular responses such as phenotypic transitions and ECM remodeling. This rationale is further described in Supplementary Section 1.

3.6. The acronym DIW should be fully spelled out upon its first occurrence.

RESPONSE: We have revised the manuscript to spell out “DIW” as “deionized water” upon its first occurrence for clarity.

3.7. There are grammatical errors throughout the manuscript. A thorough review and revision of the text are recommended to enhance its clarity and professionalism.

RESPONSE: We have carefully reviewed the manuscript and revised it accordingly.

REVIEWERS' COMMENTS

Reviewer #1 (Remarks to the Author):

I read with attention the revision of the manuscript NCOMMS-24-10590 from Kim, M. et al. "Regulation of Corneal Stromal Cell Behavior by Modulating Curvature using a Hydraulically Controlled Organ Chip Array".

The manuscript was very well improved following the reviewer's comments. In particular, the addition of dot plots for quantification of immunostaining data on Figure 3 and Figure 4 improved the demonstration of authors assertions. Moreover, the addition of RNA sequencing completed very well the qRT-PCR to examine curvature-induced transcriptomic changes across the different tested cell types. These whole-genome sequencing analyses enabled the authors to go much further in their discussion of the biological processes involved in the response of the three cell types tested to curvature. In particular, the authors were able to determine the signaling pathways involved and discuss more deeply the differences between fibroblasts, myofibroblasts, and keratocytes.

In conclusion, I consider that the manuscript has been considerably improved and can now be published.

However, I suggest a few minor corrections be made prior to publication:

p8: I do not understand the angles of $99.99^\circ \pm 3.73^\circ$ given by the authors in Figure 5a, whereas in Figure 5a no angles are given. This is surely Figure 5b. However, only angles between -90° and $+90^\circ$ are given on figure 5b... This point will need to be reviewed.

p8: the authors refer to a supplementary figure, Fic.14c, which does not exist.

Response: We thank the reviewer for this comment. To clarify the distinction, Figure 5a shows tangent-vertical angles (θ_{cross}) measured as absolute values from $0-180^\circ$, while Figure 5b presents the global orientation of individual cells, where elongation axes were normalized to -90° to $+90^\circ$. A sharper distribution centered at 0° indicates stronger global alignment.

We added indication of the angle (θ_{cross}) in Figure 5a, so that the revised figure is clear what the tangent-vertical angle is. We have revised the main text and figure legends on the text on p8 (lines 16-17 and 19-22) and p9 (lines 5-7) for high clarity.

We also update the incorrect reference to Supplementary Fig. 14c to Supplementary Fig. 6c in the revised manuscript p9 (lines 3-4).

Reviewer #2 (Remarks to the Author):

This study presents a hydraulically controlled curvature array chip to investigate how geometric cues regulate corneal stromal cell behavior. The authors characterized curvature-induced changes in proliferation, phenotype (ALDH3A1/ α -SMA), focal adhesion (vinculin/pFAK), alignment, and ECM remodeling in keratocytes, fibroblasts, and myofibroblasts. The work addresses a critical gap in corneal mechanobiology, with implications for ectatic diseases (e.g., keratoconus). The revised manuscript has addressed most of the reviewers' concerns through additional data and improved presentation. However, I have few remaining concerns and suggestions:

1. While RNA-seq identifies RhoA/ROCK and TGF- β as candidate pathways, functional validation (e.g., pharmacological inhibition) is absent. The rebuttal appropriately notes this as a future direction, but the discussion should more explicitly acknowledge this limitation.

Response: We thank the reviewer for the constructive and encouraging feedback. We have explicitly added a statement acknowledging this limitation in the discussion p21 (lines 9-16).

2. The expanded discussion on curvature modulation for wound healing/disease management (p. 21) is appreciated but remains speculative. A brief mention of specific testable strategies (e.g., curvature-adjusting implants) would enhance the impact.

Response: We thank the reviewer for this constructive suggestion. We have enhanced the discussion by including specific testable strategies for curvature modulation in clinical applications, specifically referencing intracorneal ring segments (ICRS) such as Intacs as concrete examples of curvature-adjusting implants on p22 (lines 1-10).

3. The significance of fibroblast orthogonal organization (Fig. 5) is now contextualized against prior studies using chemical/patterned substrates. However, quantitation of alignment efficiency (e.g., % cells aligned within $\pm 10^\circ$ of orthogonal axes) would strengthen claims.

Response: We thank the reviewer for this suggestion. In the revised manuscript, we have quantified fibroblast orthogonal alignment efficiency by calculating the proportion of cells aligned within $\pm 10^\circ$ of the orthogonal axis. These quantitative results have been incorporated into the *Results* section p8-9 (lines 22 and lines 1-2). In addition, we have expanded the Discussion p18 (lines 3-14) to strengthen our claims regarding alignment efficiency.

Minor revisios:

1. Terminology and grammar require refinement to ensure professionalism. For example, "DIW" should be spelled out as "deionized water" at first use. In the Abstract, "were sufficient" should be "was sufficient". In the Discussion, "e.g., "tenascin-C (TNC), a matrix glycoprotein [is] involved". A thorough language edit is recommended.

Response: We thank the reviewer for these helpful comments on terminology and grammar. We have revised the manuscript to improve clarity and professionalism.

2. Methodological transparency can be improved. Figure legends quantifying data (e.g., Fig. 3b-c) should clarify "n" (e.g., "n = number of images/cells analyzed"). Critical technical details—such as the use of a 30G syringe needle and 60 μ L/min flow rate for hydraulic actuation—

currently relegated to Supplementary Section 7, should be moved to the main Methods to enhance reproducibility.

Response: We thank the reviewer for this valuable comment. We have revised the figure legends to explicitly define $n = \text{number of analyzed images}$ and $N = \text{number of independent biological replicates}$.

Regarding the technical details, we clarify that Supplementary Section 7 describes the procedure used to generate and control the three curvatures for experiments by exploiting the self-recovery properties of PDMS. Supplementary Section 8, by contrast, details the methods used to measure and characterize the resulting curvatures. Importantly, the flow rate of 60 $\mu\text{L}/\text{min}$ mentioned in Section 8 was applied only for time-lapse volumetric characterization of curvature, not during the experimental culture conditions. To avoid confusion and enhance reproducibility, we have revised the Methods section p23 (lines 13-14 and 19-20) to clarify these distinctions.

In general, the authors have comprehensively addressed reviewer concerns through additional experiments, data re-analysis, and textual revisions. I recommend acceptance with minor revisions, after the authors further refine the text according to above comments.

Reviewer #3 (Remarks to the Author):

The authors have done an exceptional job responding to concerns raised by the reviewers and provide critical additional data in support of their original observations. In particular, the addition of the RNAseq data combined with quantification of IHC expression are strengths of the revised manuscript. The methods are highly novel and experiments conducted with rigor. This work greatly advances our understanding of how corneal curvature impacts stromal cell phenotype and gene expression.

MINOR CONCERNS:

Intro, Suggest changing "corneal haziness" to "corneal haze"

Response 1: We have replaced "corneal haziness" with "corneal haze" in the Introduction p1 (line 14).

Intro, Paragraph 3 This is an oversimplification of key growth factors in keratocyte-fibroblast-myofibroblast transformation. TGF- β also can drive keratocyte to fibroblast differentiation. I suggest reviewing this publication which nicely summarizes key growth factors in corneal wound healing within the context of different ECM environments. [10.1038/s41598-023-37776-9](https://doi.org/10.1038/s41598-023-37776-9)

Response 2: We have revised Paragraph 3 of the Introduction p1-2 (lines 20-23 and lines 1-4). We have also cited the suggested reference ([doi:10.1038/s41598-023-37776-9](https://doi.org/10.1038/s41598-023-37776-9)).

Methods, Cell Culture... What passage number of cells were used?

Response 3: The requested information has been updated in page 24, lines 16-17.

Figure 3 and Supplementary Figure 10 DAPI is used to mark nuclei (blue) in the merged images but not denoted in the figure legend.

Response 4: DAPI staining, which was used to label nuclei (blue) in the merged images of Figure 3 and Supplementary Figure 10, has now been explicitly included in the corresponding figure legends.